# Long-term perceptual priors drive confidence bias that favors prior-congruent evidence

**Marika Constant** [1,2*], **Elisa Filevich**[1,2,3], **Pascal Mamassian**[4]

**1** Humboldt-Universität zu Berlin, Faculty of Life Sciences, Department of Psychology, Berlin, Germany, **2** Bernstein Center for Computational Neuroscience Berlin, Berlin, Germany, **3** Hector Institute for Education Sciences and Psychology, University of Tübingen, Tübingen, Germany, **4** Laboratoire des Systèmes Perceptifs, Département d'Études Cognitives, École Normale Supérieure, Paris Sciences et Lettres University, CNRS, Paris, France

\* marika.constant@gmail.com

## Abstract

According to the Bayesian framework, both our perceptual decisions and confidence about those decisions are based on the precision-weighted integration of prior expectations and incoming sensory information. While it is generally assumed that priors influence both decisions and confidence in the same way, previous work has found priors to have a stronger impact at the confidence level, challenging this assumption. However, these patterns were found for high-level probabilistic expectations that are flexibly induced in the task context. It remains unclear whether this generalizes to low-level perceptual priors that are naturally formed through long term exposure. Here we investigated human participants' confidence in decisions made under the influence of a long-term perceptual prior: the slow-motion prior. Participants viewed tilted moving-line stimuli for which the slow-motion prior biases the perceived motion direction. On each trial, they made two consecutive motion direction decisions followed by a confidence decision. We contrasted two conditions – one in which the prior impacted discrimination performance, and one in which it did not. We found a confidence bias favoring the condition in which the prior influenced discrimination decisions, even after accounting for performance differences. Computational modeling revealed this effect to be best explained by confidence using the prior-congruent evidence as an additional cue, beyond the posterior evidence used in the perceptual decision. This is in agreement with a confirmatory confidence bias favoring evidence congruent with low-level perceptual priors, revealing that, in line with high-level expectations, even long-term priors have a greater influence on the metacognitive level than on perceptual decisions.

**Data availability statement:** All data, analysis scripts, and models are publicly available at https://gitlab.com/MarikaConstant/long-term-priors-in-confidence.

**Funding:** MC's work was funded by the Deutsche Forschungsgemeinschaft (DFG, German Research Foundation; https://www.dfg.de/)—337619223 / RTG2386. This work was supported by a Freigeist Fellowship (grant number 9D035-1) from the Volkswagen Foundation (https://www.volkswagenstiftung.de/) to EF. EF and PM were supported by a European Commission Doctoral Network grant "CODE" (EC MSCA-101119647; https://marie-sklodowska-curie-actions.ec.europa.eu/). The funders had no role in the conceptualization, design, data collection, analysis, decision to publish, or preparation of the manuscript.

**Competing interests:** The authors have declared that no competing interests exist.

## Author summary

Prior expectations play a critical role in shaping not only the perceptual inferences that we make, but also how confident we feel about those inferences. Bayesian confidence models capture that role, but assume priors to influence both decisions and confidence in the same way. Against this assumption, previous work has found dissociations in the influence of priors on decisions and confidence. However, that work has focussed only on high-level probabilistic priors, rather than the low-level perceptual priors that constrain our processing across many naturalistic situations. Here, we examine whether such dissociations arise under the influence of a low-level perceptual prior that naturally affects humans' perception of motion, namely, the expectation that objects move slowly. We reveal evidence for such a dissociation: prior-congruent evidence impacts confidence to a greater extent than perceptual decisions. This suggests the existence of an implicit confidence bias favoring information that confirms prior beliefs, even in the case of long-term perceptual priors.

## Introduction

Our perception of the environment is constantly compromised by uncertainties, and the Bayesian framework has become a popular way to account for how we cope with this uncertainty across a variety of cases [1–4]. According to Bayesian inference models of perception, our perceptual experience and decisions depend not only on incoming sensory information (the "likelihood"), but also on prior expectations ("priors"). These sources of evidence, often assumed to be Gaussian distributed, are combined with weights proportional to their precision to form the "posterior" probability distribution function, and the perceptual judgment is obtained by applying a decision rule on this posterior probability. More recently, these Bayesian decision models have also been extended to account for our confidence judgments about the validity of our perceptual decisions, with confidence captured as the estimated posterior probability of being correct [5–10]. This implies a role for priors in confidence as well, which has been supported empirically in both humans [11–13] and non-human animals [14].

Recent evidence suggests that priors may have dissociable impacts on perceptual decisions (first-order) and confidence judgments about those decisions (second-order). For example, priors about sensory precision, performance, or task difficulty have been shown to affect second-order judgments without changing first-order performance [15–17]. Arguably, however, this does not constitute strong evidence for a differential impact of prior information on first- and second-order decisions: These higher-order priors concerned specifically the confidence-related features of the signal, but were irrelevant for perceptual decisions. Clearer evidence comes from recent work examining how decision-relevant priors are weighted in confidence relative to their use in discrimination decisions [18]. This work found robust dissociations

between these processing levels: Prior information was used to a greater extent in subjective confidence than in the decisions themselves. This suggests that priors have a particularly strong role in confidence, over and above their effect on perceptual decisions. However, that work – along with most of the work on priors in confidence aside from a recent counter-example [19] –, manipulated priors based on probabilistic expectations that were flexibly induced in the task context and likely acted post-perceptually. One plausible explanation for these 'high-level' priors especially influencing confidence is that the abstract probabilistic information that they carry is difficult to integrate with perceptual information, but easier to integrate with confidence information. Additionally, while it may be advantageous not to rely on priors for perceptual decisions when they flexibly and rapidly change, this would not be the case for more stable, slower-updating priors. In order to test these explanations, a natural step is to ask whether the same asymmetries follow from priors that are neither difficult to integrate perceptually, nor rapidly changing, as is the case for low-level perceptual priors.

Low-level priors are often formed naturally across a breadth of life experience, such as a prior to perceive light as coming from above [20], and are sometimes therefore referred to as 'long-term' priors. Further, they have been suggested to act differently to high-level expectations in a variety of other ways, being slower to update, more cognitively impenetrable, and more context-independent [21–23]. Some have also suggested long-term priors to be implemented differently, in a more bottom-up processing stream [23,24], although there is recent counter-evidence to this argument [25]. In light of these proposed differences, it remains unclear whether the same dissociations between decisions and confidence that emerge under the influence of high-level expectations would hold for long-term perceptual priors. Here, we test this question by investigating confidence under the influence of a long-term, naturally formed prior. To do so, we focus on the 'slow-motion' prior.

It has been argued that humans have a low-level perceptual prior for motion speed to be slow, which is thought to originate from the fact that many objects we perceive in the world are stationary or move very little [26]. This has been shown to impact motion perception across different tasks and to explain a variety of biases in motion perception [26–29]. In line with Bayesian theory, the slow-motion prior influences inferences about motion particularly when sensory information is uncertain, or in other words when likelihoods are noisy. One example of uncertain motion information occurs when lines whose ends are poorly visible move through an aperture, producing the so-called "aperture effect". Under these conditions, the direction and speed of the line motion are inherently ambiguous and participants are initially biased to perceive a motion direction that is exactly orthogonal to the line orientation [30,31]. This is consistent with perceiving the slowest possible motion that can still explain the sensory information, in line with the idea that the slow-motion prior influences perceptual inference [26,32]. Sotiropoulos and colleagues found that this bias towards orthogonal motion emerged strongly in participants' motion direction decisions and was well captured by a Bayesian decision model including the slow-motion prior [32]. That work also found that the bias was reduced and even reversed by exposing participants to fast motion between test sessions, further demonstrating the impact of motion priors in this perceptual effect. Here, we build on similar stimuli as those used by Sotiropoulos et al. to investigate the effects of the slow-motion prior on perceptual decisions and confidence.

We measured confidence sensitivity and bias with the confidence forced-choice paradigm [33]. On every trial in this paradigm, participants made two consecutive perceptual decisions followed by a confidence choice regarding which of the two decisions was more likely to be correct. Each perceptual decision consisted in reporting whether a set of lines was perceived to translate in a direction that was clockwise or counterclockwise relative to a reference. We contrasted two conditions in order to examine the impact of information from the slow-motion prior on perceptual decisions and confidence judgments. One condition was tailored such that the slow-motion prior contributed strongly to whether the perceived motion direction was expected to be clockwise or counterclockwise relative to the reference, and the other condition was such that the prior was uninformative to the motion direction decision. By matching the perceptual decision rates across these conditions, we could then examine whether there was a residual confidence bias favoring one of the conditions. If so, this could suggest that confidence uses the prior information differently than the first-order decisions do. Further, to quantify this effect and better understand the mechanisms underlying the use of perceptual priors in confidence, we fit and compared several models of confidence choices under the influence of priors.

## Results

On every trial of the confidence forced-choice task, participants decided about the motion direction of a set of parallel lines moving through a circular aperture (Fig 1A) on two consecutive and independent stimulus intervals. The preferred motion direction due to the slow-motion prior was orthogonal to the orientation of the moving lines. On the outside of the circular aperture, two abutting arcs, each subtending 90°, served as choice regions – one orange, one blue – and defining the

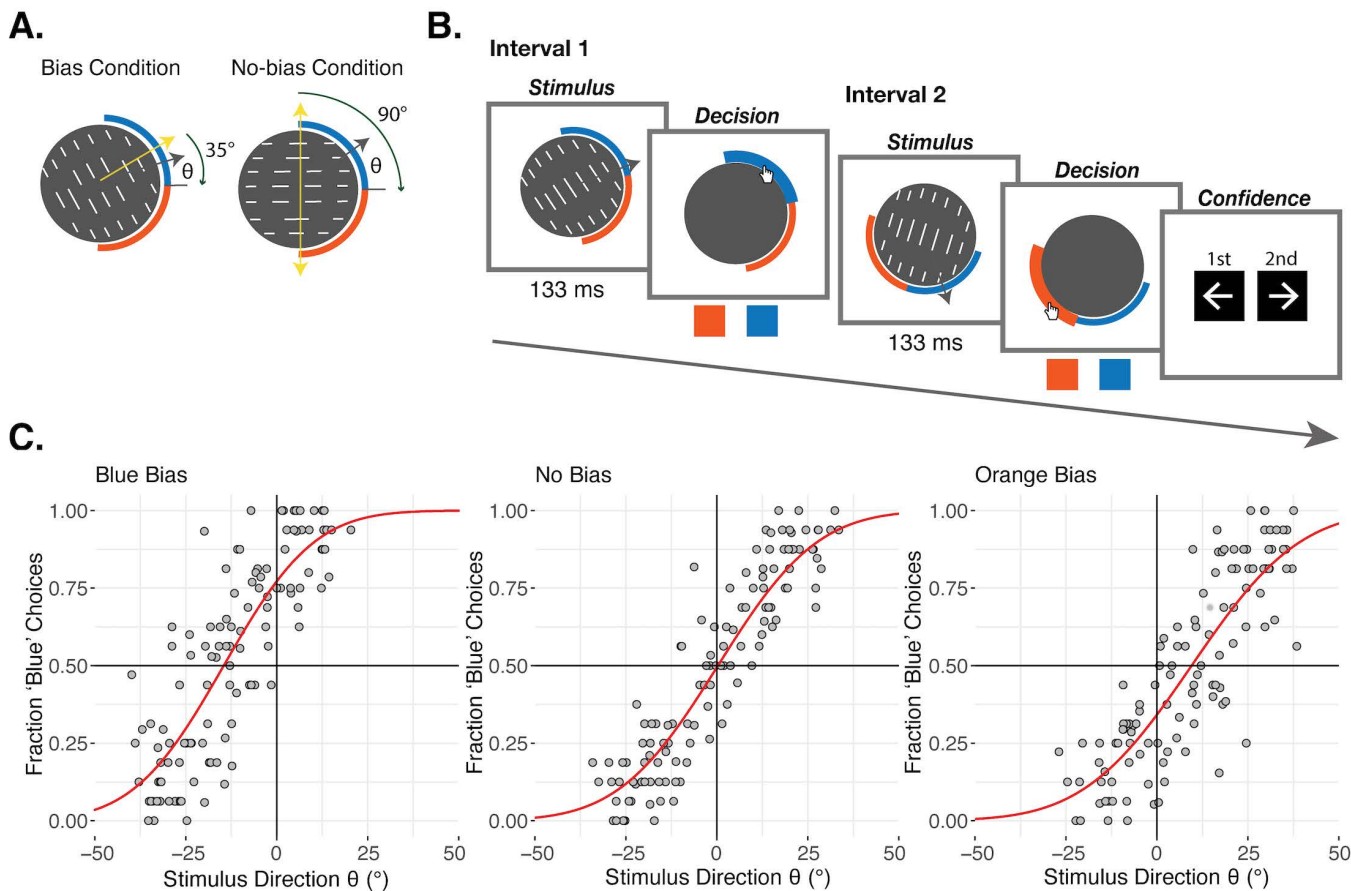

**Fig 1. Task and manipulations. A. Conditions and sketch of stimuli.** The stimuli consisted of a set of lines moving through a circular aperture, shown at low contrast and for a short duration. The yellow arrows indicate the preferred motion directions consistent with the prior, orthogonal to the lines' orientation. In the Bias condition, the angle between this prior and the reference was 35°, creating a decision bias favoring the color in which the prior falls. In the No-Bias condition, that angle was 90°, such that we expected no prior-induced decision bias favoring either color. The gray arrows indicate the true motion direction on an example stimulus. The difficulty of the stimuli (and mean of the likelihood) was controlled by the angle, θ, between this motion direction and the reference. A larger θ was required in the No-Bias condition to reach the same choice rates of seeing a motion direction in the blue region, since the prior favored blue and orange decisions equally. **B. Experimental task.** A trial of the confidence forced-choice task included two stimulus intervals, in which participants saw a moving stimulus and made an orange/blue decision about the motion direction of the lines, followed by a confidence forced-choice about which of their perceptual decisions they were more confident of being correct. **C. Manipulation check.** Pooled results from the adaptive staircase (ASA) procedure in which many different θ values were tested in their effect on perceptual decisions in both conditions before the full confidence task. Each gray point corresponds to the responses for a given stimulus intensity level from one participant. θ values towards the blue region are encoded as positive, and towards the orange region are encoded as negative. Red psychometric functions show the fit cumulative normal distribution function capturing the relationship between the θ and the fraction of blue choices. In the No-Bias condition (middle panel), there is no choice bias for orange or blue. In the Bias condition, when the prior is 35° away from the reference in the blue region (left panel), the function is shifted to the left, showing a successfully induced choice bias favoring blue. When the prior is 35° away from the reference in the orange region (right panel), the function is shifted to the right, showing a successfully induced choice bias favoring orange.

'reference' at their separation (Fig 1A). In each interval participants made a decision about whether the motion direction was towards the displayed blue or orange region. Participants then made a confidence forced-choice on every trial about whether they were more confident about being correct in the first or second interval (Fig 1B), thereby removing issues of confidence biases that come with subjective ratings [34]. We created two conditions. In the Bias condition, the preferred direction – orthogonal to the line orientation – fell within either the orange or blue region, thus creating a decision bias (Fig 1A). In the No-Bias condition, the preferred motion direction(s) fell exactly 90° away from the reference and hence there was no bias favoring either decision. In this way, we could compare the No-Bias condition in which decision performance was determined entirely by the incoming sensory information (the likelihood) against the Bias condition, in which the slow-motion prior influenced performance. The difficulty of the stimuli was controlled by the angle, θ, subtended by the true motion direction and the reference, and this formed the mean of the likelihood distribution. In each condition, we used four θ values, targeting 0.15, 0.35, 0.65, and 0.85 probabilities of choosing blue (P('Blue')). The Bias condition could be set to induce a bias favoring either orange or blue, by either placing the preferred motion direction in the orange or blue region. However, we wanted to focus on the case in which the Bias condition required *less* stimulus information than the No-Bias condition, due to the contribution of the prior. So, in the experiment, we did not include cases in which the bias worked against the stimulus. Therefore, when inducing a blue bias (with the preferred motion direction in the blue region), we only included θ values targeting 0.65 and 0.85 P('Blue'), and when inducing an orange bias (with the preferred motion direction in the orange region), we only included θ values targeting 0.15 and 0.35 P('Blue').

## Manipulation check

We predicted that the slow-motion prior would bias participants' perceptual decisions in the Bias condition, and hence that the Bias condition would require a smaller θ angle in order to lead to the same decision rates as the No-Bias condition. This effect would serve as a basic manipulation check to ensure that the conditions and slow-motion prior had the intended impact. Additionally, it was necessary to quantify this effect in each participant before the experiment such that we could choose the θ values that would be expected to lead to matched perceptual decision rates. To do this, before the main task we ran a staircasing procedure which sampled different θ values in both conditions and then fit psychometric functions to these data, including the bias and sensitivity per condition. The fitted psychometric functions across all pooled participants are shown in Fig 1C, and the bias effect of interest was verified in each individual participant (Fig A in S1 Text). Motion towards the orange region was encoded with negative θ's and blue region with positive θ's. We found a clear bias effect, such that the psychometric was shifted leftward when the preferred motion direction from the prior was 35° from the reference in the blue region and rightward when it was 35° from the reference in the orange region, compared to the No-Bias condition (Fig 1C). The point of subjective equality (PSE) when there was a blue bias was -14.57°, meaning that when the line motion direction was 14.57° into the orange region, participants nonetheless chose orange and blue at equal rates. In contrast, the PSE when there was an orange bias was 9.53°, and the PSE in the No-Bias condition was 0.30°. Together, this suggests that the bias manipulation worked as planned.

## Motion-direction decisions

After running the staircasing procedure, we chose the values of θ angles that corresponded to a 0.15, 0.35, 0.65 and 0.85 expected probability of choosing blue for each participant. This procedure was followed both for the No-Bias condition and for the Bias condition (for a total of eight different θ values). In the Bias condition, the values corresponding to a 0.15 and 0.35 P('Blue') were taken from cases in which the slow-motion prior induced an orange bias (the lines were oriented to have the orthogonal direction in the orange region), and values corresponding to a 0.65 and 0.85 P('Blue') were taken from cases in which the prior induced a blue bias (the lines were oriented to have the orthogonal direction in the blue region). Overall, this meant that we chose stimulus levels that were expected to lead to matched perceptual decision rates between conditions. Crucially, however, the perceptual decision rates in the Bias condition would be due in part to the

slow-motion prior; whereas the perceptual decision rates in the No-Bias condition would be solely driven by the stimulus evidence. These four θ values in each condition were paired in all possible combinations for the two intervals of the confidence forced-choice task, except for pairs repeating the identical θ in the same condition.

To check that the perceptual decision rates were matched overall across the two conditions, we ran a logistic mixed effects model on Perceptual Decision including Expected P('Blue') (as an index of the stimulus difficulty), Condition (Bias vs No-Bias), and their interaction as fixed effects, as well as by-participant random intercepts. If the chosen θ values from the staircasing procedure failed to produce matched perceptual decision rates between conditions, this would generate an interaction between Expected P('Blue') and Condition. Indeed, a significant interaction, $\chi^2(3)=302.97$, $p<0.001$, $BF_{10}=1.19\times10^{60}$, revealed that the decision rate per stimulus setting additionally depended on the condition, making our analysis more complex. The perceptual decision rates were more extreme in the Bias condition (Fig 2A), with higher odds of choosing blue when the stimulus was blue (Expected P('Blue') of 0.65: odds ratio of the Bias to the No-Bias condition ($OR_{Bias/No-Bias}$) = 1.55, 95% confidence interval (CI) [1.32, 1.81], $Z=8.75$, $p<0.001$; Expected P('Blue') of 0.85: $OR_{Bias/No-Bias}=1.13$, 95% CI [0.92, 1.38], $Z=1.81$, $p=0.071$), and lower odds when the stimulus was orange (Expected P('Blue') of 0.15: $OR_{Bias/No-Bias}=0.54$, 95% CI [0.42, 0.69], $Z=-7.91$, $p<0.001$; Expected P('Blue') of 0.35: $OR_{Bias/No-Bias}=0.49$, 95% CI [0.41, 0.58], $Z=-13.43$, $p<0.001$), compared to the No-Bias condition, although this contrast was not significant for the strongest blue setting. The more extreme perceptual decision rates in the Bias condition suggest that the bias effect was stronger during the main experimental trials, compared to what was calibrated in the staircasing procedure. To quantify this, we estimated the strength of the bias effect in the experiment by fitting a cumulative normal distribution function with one symmetrical bias level (+/- $\mu_{Bias}$) for the orange and blue directions relative to the No-Bias condition, which we fit separately as a baseline (Fig 2A). The sensory sensitivity was also allowed to differ between conditions, as it was possible that the influence of the prior impacted the overall sensitivity. This was done for each participant and the resulting mean $\mu_{Bias}$ across participants was 20.81° (Fig 2A) with a SEM of 2.68° (individual fits in Fig B in S1 Text), indeed indicating a stronger bias effect than in the staircasing (Fig 1C). Hence, we needed to account for this residual difference in perceptual decision rates in our analysis when testing for a confidence bias.

## Confidence forced-choices

We sought to explore whether, after accounting for any residual difference in orange/blue decision rates, there is still a confidence bias favoring one of the conditions. If so, this could indicate a different influence of the prior on perception and confidence, which could then be further explored using Bayesian confidence modeling. Because the orange/blue decision rates were not perfectly matched across conditions, and following our pre-registered plan for this potential situation, we used a non-parametric approach to assess confidence bias, which accounted for potential mismatches in perceptual decision rates. For this analysis we excluded trials with the same condition in both intervals. We took the difference in logit transformed orange/blue response rates for the preferred stimulus between conditions for every stimulus setting pair for every participant, following previous work [35]. For example, for the stimulus settings expected to lead to a 0.85 probability of choosing blue in each condition, we took the observed rates of choosing blue, took their log-odds to transform them to the (-∞,∞) domain, and took the difference between them. A difference of zero then indicated matched perceptual decision rates, positive differences indicated more extreme decision rates in the Bias condition, and negative differences indicated more extreme decision rates in the No-Bias condition. Then, regardless of whether these values are shifted positively overall, as we expect due to the result of more extreme perceptual decision rates in the Bias condition, we can fit a cumulative normal distribution function to the relationship between these transformed response proportion differences and the probability of choosing the Bias condition as the more confident interval. If the PSE is at 0, this indicates that when participants have matched perceptual decision rates they also have matched confidence choice rates between conditions, suggesting no confidence bias. If, however, the PSE is shifted away from 0, this indicates a confidence bias. We found a clear negative shift of this function, fit to the pooled participants, with a PSE of -1.47 (SEM = 0.24 from performing this

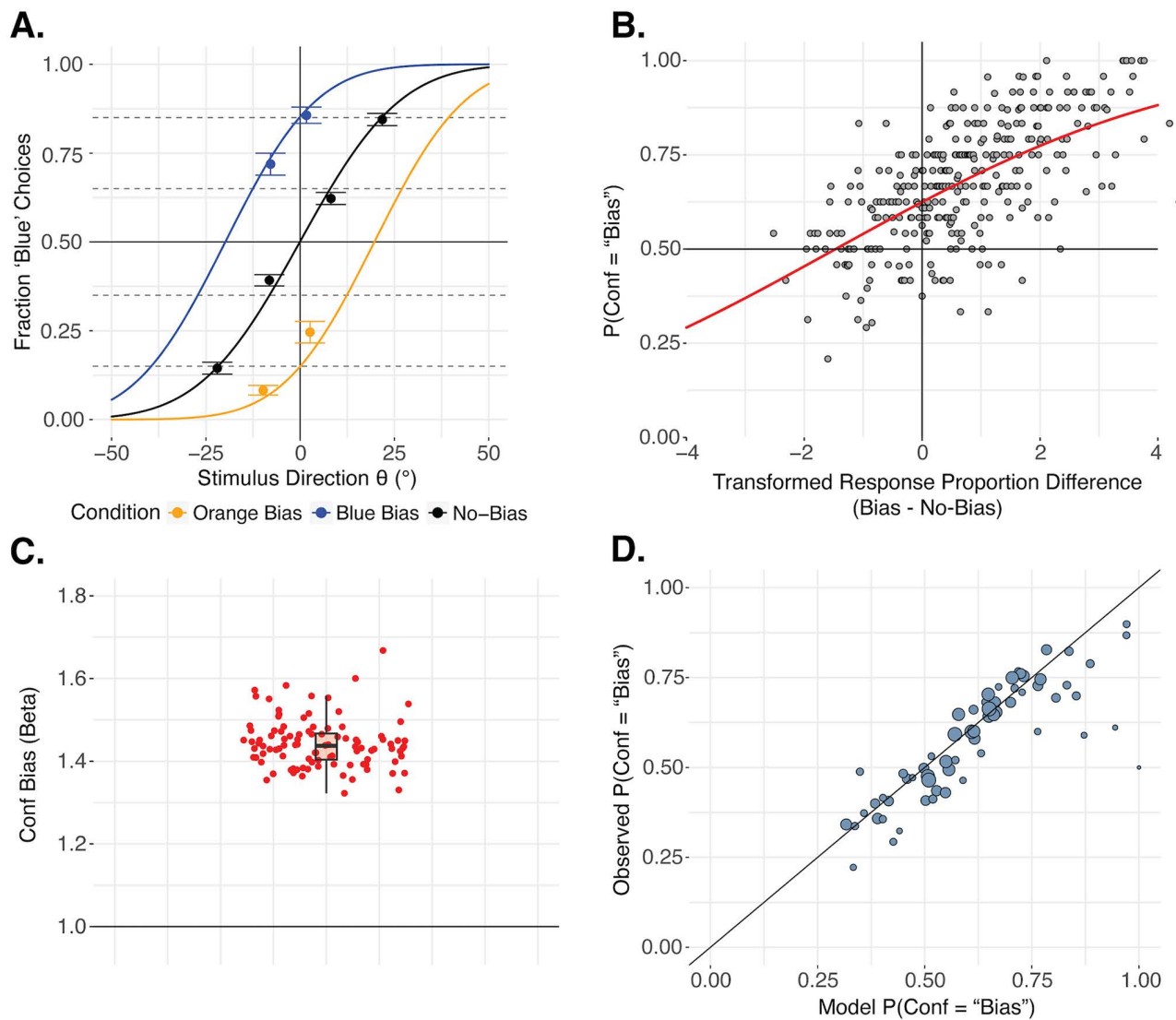

**Fig 2. Motion-direction and confidence decision results. A. Motion-direction decisions.** Choice results for the first-order orange/blue decisions in each condition. The four θ values in each condition were targeting 0.15, 0.35, 0.65 and 0.85 probabilities of choosing blue (P('Blue')). In the Bias condition, this consists of θ values targeting 0.65 and 0.85 P('Blue') when inducing a blue bias (blue points) and targeting 0.15 and 0.35 P('Blue') when inducing an orange bias (orange points). The targeted perceptual decision rates are shown with horizontal dashed gray lines. Although we targeted these matched rates between conditions, the observed perceptual decision rates were less extreme in the No-Bias condition than in the Bias condition. Error bars reflect the SEM across participants. The psychometric functions were fitted to each individual participant to capture the strength of the bias from the slow-motion prior relative to the No-Bias baseline, and to capture possible differences in sensitivity between conditions. The functions shown here reflect the mean fit bias (μ) and noise (σ) values across participants (individual fits in Fig B in S1 Text). The mean $\mu_{Bias}$ was 20.81, mean $\mu_{No-Bias}$ was 0.44, mean $\sigma_{Bias}$ was 18.50, and mean $\sigma_{No-Bias}$ was 20.46. **B. Non-parametric confidence bias results.** On the x axis is the difference in the logit transformed rates of choosing the expected color between the Bias and No-Bias condition. Each gray point represents this difference for one pair of stimulus settings assigned to the two intervals of a confidence pair (16 points per participant). On the y axis is the probability of choosing the Bias condition interval as the more confident interval. When there is no difference in perceptual decision response proportions between conditions (x = 0), and the perceptual difficulties are therefore matched for that pair of settings, we expect equal confidence choices between conditions. If this is not the case, this indicates a confidence bias. The red psychometric function captures the fit cumulative normal distribution function to the relationship between this response proportion difference and the confidence choice rates from the pooled data. The PSE is shifted leftward to -1.47 (SEM = 0.24 from performing this analysis in individual participants, see Fig C in S1 Text), indicating a confidence bias favoring the Bias condition. **C. CFC-model confidence bias results.** We fit the *cfc-model* [33] across 100 bootstrapped runs and extracted the fit confidence bias parameter β from each. Each red point corresponds to one fit β parameter result. The boxplot shows the median, interquartile range (IQR) with hinges showing the first and third quartiles, and vertical whiskers stretching to most extreme data point within 1.5*IQR from the hinges. The horizontal black line at y = 1 shows the expected β if there was no confidence bias. **D.**

**Quality of CFC-model fit.** Predicted against observed confidence choice rates across all participants. Each point shows a pair of stimulus settings with one interval in each condition and a given set of two orange/blue responses, and the size of the point reflects the number of trials. The closer the points are to the x=y line, the better the model is at modeling confidence choice rates.

analysis in individual participants, see Fig C in S1 Text), suggesting a confidence bias favoring the Bias condition (Fig 2B). When perceptual decision rates are matched (transformed response proportion difference=0), the fit psychometric function predicts the participants to still choose the Bias condition as the more confident interval at a rate of 0.62. Note that this reflects the degree to which confidence was *more* biased towards the Bias condition than perceptual decisions, rather than the overall degree to which the slow-motion prior influenced confidence.

### Quantifying confidence bias: CFC-model

To further quantify this confidence bias, we fit the *cfc-model* developed in previous work [33] for use with the confidence forced-choice paradigm. This model includes a confidence bias term, $\beta$, which, for a single task or condition, scales the estimated sensory sensitivity of a participant, such that $\beta = 1$ captures correctly estimated sensitivity, $\beta > 1$ captures an overestimation of sensitivity – indicating overconfidence –, and $\beta < 1$ captures an underestimation of sensitivity – indicating underconfidence. With two different conditions, the model fits $\beta$ as the ratio of biases between them (one can only estimate whether participants are over- or under-confident in one condition relative to the other). In our case, we chose the 'No-Bias' condition as baseline, so a fit $\beta$ larger than 1 would indicate an overconfidence in the 'Bias' condition. We fit the model to the pooled normalized data from all participants across 100 bootstrapped runs, which revealed the mean fit $\beta$ to be 1.44 (SEM=0.006; Fig 2C). The quality of the model fit is shown in Fig 2D. These results, in agreement with the non-parametric analysis above, indicate a confidence bias favoring the Bias condition. While this model can be used to naively quantify a confidence bias relative to first-order perceptual decision rates, it does not provide an account of the computations occurring in our design, under the influence of the slow-motion prior. To investigate these computations, in exploratory analyses we consider a Bayesian decision and confidence model. We use this model as a normative benchmark, specifically focussing on a Bayesian use of the slow-motion prior. We then explore several alternative generative models that may help capture deviations from this Bayesian baseline, in light of the observed confidence bias.

### Bayesian model

In the Bayesian model, we first need to define the prior to represent the preferred motion directions that are orthogonal to the lines' orientation. Even though we relied on the slow-speed prior to bias one condition, this prior has a direct impact on perceived direction [30], and since we are measuring perceived direction rather than speed, we directly model the prior in terms of the preferred motion directions. This motion direction prior is implemented as the sum of two Gaussians of equal variance, whose means are separated by 180° (Fig 3A). In the No-Bias condition, these means are 90° away from the reference in either direction. In the Bias condition, one mean is 35° away from the reference in the bias direction, and the other is 145° away from the reference in the opposite direction. This latter component is outside the allowed decision region and thus will have a negligible effect in our model (see below). The likelihood is modelled as a single Gaussian centered around the internal signal generated from the stimulus with added internal noise. These combine to form the posterior distribution (Fig 3A). Orange/blue perceptual decisions in this model are based on the area under the posterior on either side of the sensory reference within the allowed decision region (from -90° to +90°). When this area is larger on the negative side, an "orange" decision is made, and when it is larger on the positive side, a "blue" decision is made (Fig 3A–B). Following the literature capturing Bayesian confidence as the estimated posterior probability of being correct [5–10], confidence in the decision is then equal to the proportion of the total area (within the allowed decision region) that falls on the chosen side (thus varying between 0.5 and 1). The confidence forced-choice is based on whichever of these

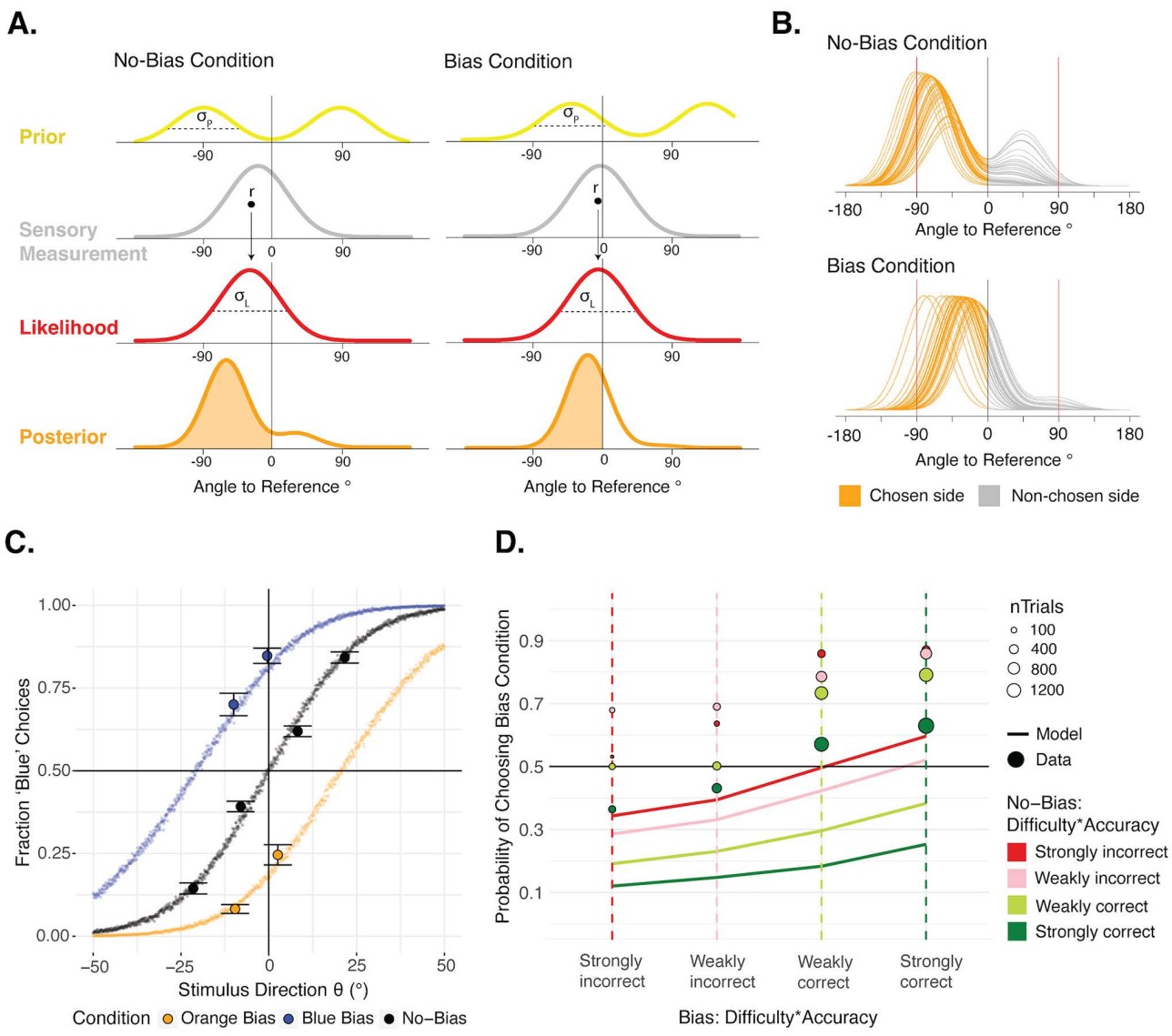

**Fig 3. Bayesian model. A. Sketch of Bayesian model.** The prior on perceived motion direction is a sum of two Gaussians of equal standard deviation ($\sigma_P$), with means at the orthogonal motion directions (+/-90° from the reference in the No-Bias condition and either 35°/-145° or -35°/145° from the reference in the Bias condition). The likelihood comes from the internal signal, *r*, sampled from the stimulus with added internal noise ($\sigma_L$). The prior and likelihood are combined to form the posterior distribution. The shaded orange area under the posterior leads to the first-order decision and confidence. **B. Posteriors per condition from Bayesian model.** Posterior distributions from the Bayesian model across 50 samples of an internal signal from the stimulus to form the likelihood, only including samples leading to an orange decision. These reflect posteriors across different trials of the same stimulus intensity and condition, which led to the same perceptual decision. The confidence in these trials is captured by the ratio of area under the orange part of the posterior (chosen side) to the area under the gray part of the posterior (non-chosen side), between the red vertical lines (allowed decision region). The prior tends to pull the maximum of the posterior in the No-Bias condition away from the reference towards -90°, so the ratio of area under the orange versus gray parts of those posteriors tends to be higher, suggesting higher average confidence in the No-Bias condition despite matched choice rates. **C. Modeled motion-direction decision results.** Results from the fit Bayesian decision model, simulated across 1001 stimulus intensities ranging continuously (in steps of 0.1) from -50 to 50 using the best-fitting $\sigma_P$ and $\sigma_L$ parameters for each participant, against the data. **D. Modeled confidence choice results.** In two consecutive intervals, a Bias and a No-Bias stimulus were presented (or in the reverse order). Because confidence refers to an estimate that a perceptual decision is correct, we consider each possible combination of difficulty and perceptual decision accuracy. 'Strongly correct' refers to trials with either an extreme blue stimulus (Expected P('Blue') of 0.85) and a 'Blue' perceptual decision, or an extreme orange stimulus (Expected P('Blue') of 0.15) and an 'Orange' perceptual decision. 'Strongly incorrect' refers to trials with those extreme stimulus settings but opposing perceptual decisions. The 'Weakly correct' and 'Weakly incorrect' categories follow the same patterns but with the more difficult stimuli: Expected P('Blue') of 0.35 or 0.65. Results split further by orange versus blue stimulus settings and perceptual decisions are shown in Fig D in S1 Text. The x-axis captures this combination of difficulty and perceptual decision accuracy in the Bias condition interval —also indicated by thin, colored, dashed vertical lines—. The

combination of difficulty and perceptual decision accuracy in the No-Bias condition interval is captured by color. The colors correspond to the vertical lines of the Bias condition, such that where a colored point intersects a dashed vertical line of the same color, there is the same degree of difficulty and perceptual decision accuracy across the two conditions, and an unbiased confidence observer would choose both conditions at equal rates for their confidence. The y-axis captures the confidence choice rates favoring the Bias condition. Simulated confidence choice rates from the Bayesian model are shown by the solid lines and the observed confidence data are shown by circles whose size reflects the number of trials. The overall shift upward of the confidence data points compared to the model predictions indicates that observed confidence choices favored the Bias condition more often than would be expected of the Bayesian model, which actually predicts a bias in the opposite direction.

confidence values is larger across two intervals. The free parameters of this Bayesian model are the standard deviation of each of the two Gaussians that form the Gaussian mixture prior, $\sigma_P$, and the standard deviation of the likelihood, $\sigma_L$, that represents the sensory uncertainty. These were fit individually for each participant to find the parameters that could best explain their orange/blue decisions. We used a simplification to deal with computational complexity and considered the prior in the Bias condition to be a single Gaussian centered around 35° towards the bias direction. Since most of the other Gaussian fell well outside the allowed decision region, this component had a negligible impact. The results from these model fits are shown in Fig 3C by simulating from the full unsimplified model, showing that the fit $\sigma_P$ and $\sigma_L$ parameters can still well account for motion-direction decisions.

We then used this fit model and investigated the confidence forced-choice patterns that would be expected of a Bayesian observer given the first-order perceptual decision rates that we found (Fig 3D, solid lines), and compared them to the observed confidence patterns (Fig 3D, data points). For the analyses based on the Bayesian model, we removed all trials with the same condition in both intervals. Because of the confidence forced-choice task structure and the fact that confidence refers to an estimate that a perceptual decision is correct, we should distinguish confidence choices depending on the stimulus difficulty and perceptual decision (blue and orange) of each of two intervals on a given trial. For example, if the Bias condition interval is easy and the No-Bias condition interval is difficult, there is a high probability of a confidence choice favoring the Bias condition. Likewise, if the Bias condition perceptual decision is correct and the No-Bias condition perceptual decision is incorrect, there would be a tendency for the confidence choice to favor the Bias condition with all else equal. Hence, the confidence choice rates in Fig 3D (y-axis) can be viewed as a function of the combination of stimulus difficulty and perceptual decision accuracy in each condition, with the Bias condition on the x-axis and the No-Bias condition in color. 'Strongly correct' trials refer to those with extreme θ values targeting a 0.85 rate of either 'Blue' or 'Orange' perceptual decisions, and on which the participant made the corresponding, expected 'Blue' or 'Orange' perceptual decision. 'Strongly incorrect' trials refer to those with the extreme θ values but on which the participant made the opposing perceptual decision. 'Weakly correct'/'Weakly incorrect' trials follow the same pattern but refer to trials with the more difficult θ values targeting a 0.65 rate of either 'Blue' or 'Orange' perceptual decisions. When the combination of difficulty and perceptual decision accuracy is matched across conditions (when the color of the No-Bias condition intersects the vertical dashed line of the same color of the Bias condition), an unbiased confidence observer would have a confidence choice rate of 0.5. Instead, observed confidence choice rates are shifted towards the Bias condition overall, suggesting the same confidence bias seen in Fig 2B. However, it is important to note that the stimulus difficulties shown are grouped by *targeted* perceptual decision probability, and we were not perfectly successful at achieving these decision rates. So, the shift of the data points towards the Bias condition cannot be interpreted in isolation, as it may reflect the failure to match perceptual decision rates. Instead, the data must be compared to the model predictions in these Figs. The raw confidence choices are also in striking contrast to the predicted confidence choice rates of the Bayesian model, shown in the solid lines of 3D, which actually show a confidence bias in the opposite direction.

The expected rates of confidence choices favoring a condition depend not only on stimulus and perceptual decision accuracy, but on the relative strengths of the *posteriors* in each condition, and therefore on the use of the prior. In the Bayesian model, because of the shape of the Gaussian mixture priors, the distribution of posteriors across different samples of the stimulus (or in other words across trials) is quite different between the conditions (Fig 3B). In the No-Bias

condition, the maxima of these posteriors are pulled towards the peaks of the prior at either + or -90°, so a large pro-portion of the area is distributed relatively far from the sensory reference on the chosen side (Fig 3B). Having a higher proportion of the area on the chosen side like this leads to higher confidence, and therefore the Bayesian observer will on average have higher confidence in the No-Bias condition despite matched perceptual decision rates. This pattern is shown clearly in the model simulations of the Bayesian confidence observer given the first-order decision results (Fig 3D). This suggests that, if our participants used the information from the slow-motion prior in a Bayesian optimal manner for their confidence, there would have been a confidence bias favoring the No-Bias condition, which is the opposite of what is seen in the data. So, we next explore some alternative confidence models that might better explain the patterns found.

## Distance-to-Criterion confidence model

In the Bayesian confidence model, confidence is based on the area under the full Bayesian posterior. However, another possibility for computing confidence is to base it off of a single sample from the posterior as a heuristic, in which case the brain does not need to perform the full integration (see, e.g., [36]). Such posterior sampling is still debated in the litera-ture, but there is some evidence in support of it [37]. In this model, confidence is proportional to the distance between the posterior sample and the sensory reference, or 'criterion' (Fig 4A). Samples further from the criterion very clearly support that perceptual decision and lead to higher confidence, whereas samples close to the criterion indicate uncertain deci-sions and lead to lower confidence. We also assume that confidence would never *oppose* the perceptual decision, so we only considered samples for confidence that fell on the same side of the criterion as the perceptual decision. This confi-dence model has no free parameters beyond the first-order sensitivity parameters fit to perceptual decisions and used in all models. We simulated confidence forced-choices from it to investigate whether it could better account for the data. This revealed even more extreme predictions of a confidence bias favoring the No-Bias condition than in the Bayesian model,

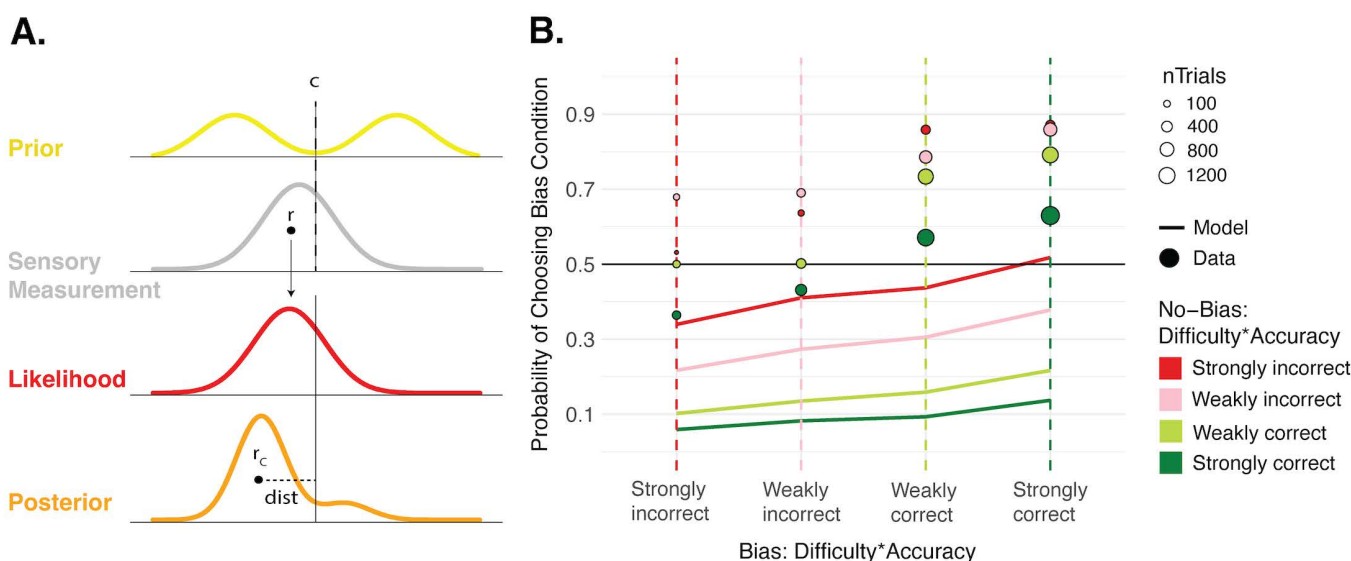

**Fig 4. Distance-to-Criterion confidence model. A. Sketch of Distance-to-Criterion model.** In the Distance-to-Criterion model, the prior, likelihood, and resulting posteriors are the same as in the Bayesian model. Motion-direction decisions also work identically. However, at the confidence level, instead of computing the full Bayesian posterior probability of being correct, this model bases confidence on the absolute distance, *dist*, between a single sample from the posterior, $r_c$, and the reference, or criterion **(c)**. Larger distances lead to higher confidence and smaller distances lead to lower confidence. **B. Confidence predictions from Distance-to-Criterion model against data.** Confidence forced-choice predictions simulated from the Distance-to-Criterion model (solid lines) against the confidence data (points) in the same format as in Fig 3D. Like in the Bayesian model, the solid lines well below the data points show that this model predicts more confidence choices favoring the No-Bias condition, unlike what is seen in the data.

suggesting this model could only poorly capture the confidence data (Fig 4B). Beyond posterior sampling, other possible heuristics include basing confidence on the mean of the posterior distribution, but this is more computationally costly, and does not substantially change the results (Fig E in S1 Text).

## Decoupled Prior confidence model

The Bayesian and Distance-to-Criterion models both base confidence on the same posterior as the first-order decisions, and fail to capture the confidence bias seen in the data favoring the Bias condition. It is possible that the confidence data can be better explained by using a weaker likelihood relative to the prior for computing confidence than for perceptual decisions, leading to a stronger influence of the prior in confidence, similar to the pattern found in previous work [18]. In other words, the strength of the prior that is used in confidence may decouple from that used in the perceptual decision. In the Decoupled Prior (DP) model, the variance of the likelihood is scaled by a multiplicative factor $w$ during the confidence computation to capture a relative over- or underestimation of the likelihood precision and hence an over- or underweighting of the likelihood information relative to the prior (Fig 5A). When $w = 1$, the likelihood variance is correctly estimated, the relative precision-weighting is optimal, and this model is identical to the Bayesian confidence model. When $w > 1$, the likelihood variance is overestimated and therefore it has a weaker effect on the computation of confidence, and the prior is relatively overused in confidence. When $w < 1$, the likelihood variance is underestimated and therefore it has a stronger effect and the prior is relatively underused in confidence. In essence, this model is a naive implementation of the confidence bias parameter in the Confidence-Forced-Choice model [33] where large values of $w$ correspond to small values of $\beta$. We fit this model with its free parameter $w$ to the group data using a maximum likelihood estimation approach. This revealed

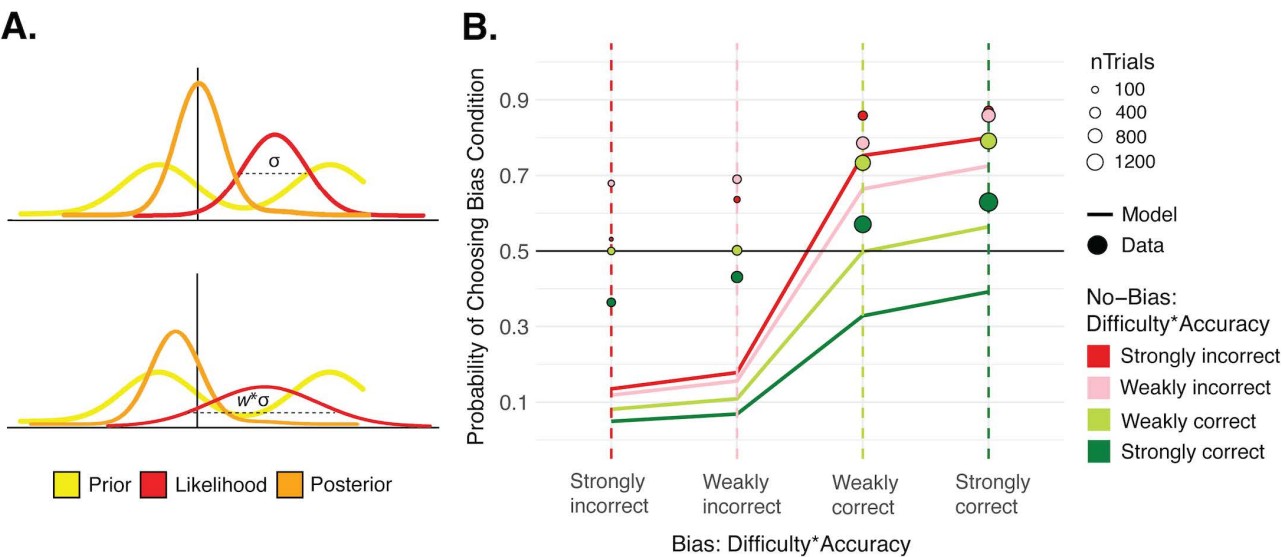

**Fig 5. Decoupled Prior confidence model. A. Sketch of Decoupled Prior model.** In the Decoupled Prior model, confidence is computed the same way as in the Bayesian model, by taking the area under the posterior distribution on the chosen side. However, in forming this posterior, participants over- or underestimate the precision of the likelihood relative to the prior, leading to the prior being relatively over- or underused in the confidence computation. The over- or underestimation of the likelihood precision is captured by a scaling term $w$ that acts as a multiplicative factor on the variance of the likelihood. When this value is 1, the variance is correctly estimated and this model is the same as the Bayesian model. When $w < 1$, the prior is relatively underused and when $w > 1$ the prior is relatively overused in confidence. **B. Confidence predictions from fit Decoupled Prior model against data.** Confidence forced-choice predictions simulated from the fit Decoupled Prior model (solid lines) with the best-fitting $w = 1.54$, against the confidence data (points). Although the model fits the data better than the Bayesian model, it still underpredicts the confidence bias favoring the Bias condition compared to the data.

a best-fitting parameter of $w = 1.54$, indicating that the prior is overused relative to the likelihood in confidence. This is in agreement with the results of previous work which also found an overweighting of prior information at the confidence level using a similar model implementation [18]. It may also align with work finding that people base inference on an underestimation of their own uncertainty [38], though that work focussed only on likelihood variance, assuming uninformative priors.

Simulations from this fit model are shown in Fig 5B. These demonstrate the ability of this model to better capture the confidence bias favoring the Bias condition than the Bayesian and Distance-to-Criterion models. However, this model predicts confidence to be particularly high in the Bias condition when the perceptual decision is in agreement with the prior, and to be particularly low when the perceptual decision goes against the prior. This leads to the prediction that confidence choices will favor the Bias condition when the Bias condition perceptual decision agrees with the bias direction. But, the model also predicts that confidence choices will favor the No-Bias condition when the Bias condition perceptual decision goes against the bias direction, which cannot explain the data for these cases (Fig 5B, the red and pink vertical dashed lines).

### Prior-Congruent Evidence (PCE) confidence model

From the previous models, it seems that confidence forced-choices deviate from the Bayesian model and are biased towards the condition that is more driven by the prior, but not simply by overweighting the prior (or downweighting the likelihood) in the integration to form the posterior. This suggests that confidence may instead be biased towards the prior by an additional source of evidence, and a plausible candidate for this additional cue is the prior-congruent evidence (PCE) available in the line-motion stimulus itself. The prior-congruent evidence refers to the component of the stimulus motion speed that is projected on the axis orthogonal to the lines' orientation. It is not tied to the orange/blue decision made, and is separate from the posterior evidence on which that decision is based. In an extreme case, confidence could be based *solely* on this PCE, ignoring the posterior evidence entirely and dissociating from the perceptual decision. This is the case illustrated by the PCE confidence model. Confidence here was directly proportional to the length of the prior-congruent vector component of the sensory evidence (a noisy measurement of the stimulus), as shown in Fig 6A. There were no additional free parameters for confidence in this model. Because there tended to be more of this prior-congruent evidence in the Bias condition (when line motion directions were closer to the preferred orthogonal direction), this model predicts a confidence bias favoring the Bias condition (Fig 6B). However, the predicted confidence bias is stronger than what we see in the data. This is not surprising, as this model serves primarily to illustrate the influence of the PCE cue. But, it is included for completeness as it is still a possible confidence strategy, albeit an extreme one.

### Weighted Posterior and Prior-Congruent Evidence (WPPCE) confidence model

In the WPPCE model, the PCE acts as an additional confidence cue, but not the sole information on which confidence choices are based. Confidence is based on a weighted combination of the Bayesian posterior evidence (BPE), which drives the first-order decisions, as well as the degree of prior-congruent evidence (PCE). The influence of each of these two evidence sources is determined by the weighting parameter, α, such that confidence is proportional to the sum of (1-α)*BPE and α*PCE. Higher α values then lead to more contribution from the PCE. When α = 1, the PCE is the only cue to confidence and this is identical to the PCE model, and when α = 0, the BPE is the only cue to confidence and this is identical to the Bayesian model. We fit this free weighting parameter to the group data using a maximum likelihood estimation approach, and found that a value of α = 0.34 could best explain the data. Because the BPE and PCE use different units, it is difficult to interpret the raw α value. However, simulations from this fit model are shown in Fig 6C, and can capture the data well qualitatively, showing a similar confidence bias for the Bias condition. In an exploratory analysis, we also examined whether the strength of deviation away from Bayesian confidence, as captured by the weight of the PCE (α), correlates to the strength of the prior at the perceptual level. While it did not reach significance, we did find a trend towards a positive relationship between these variables (Fig F in S1 Text), which could be explored in future research.

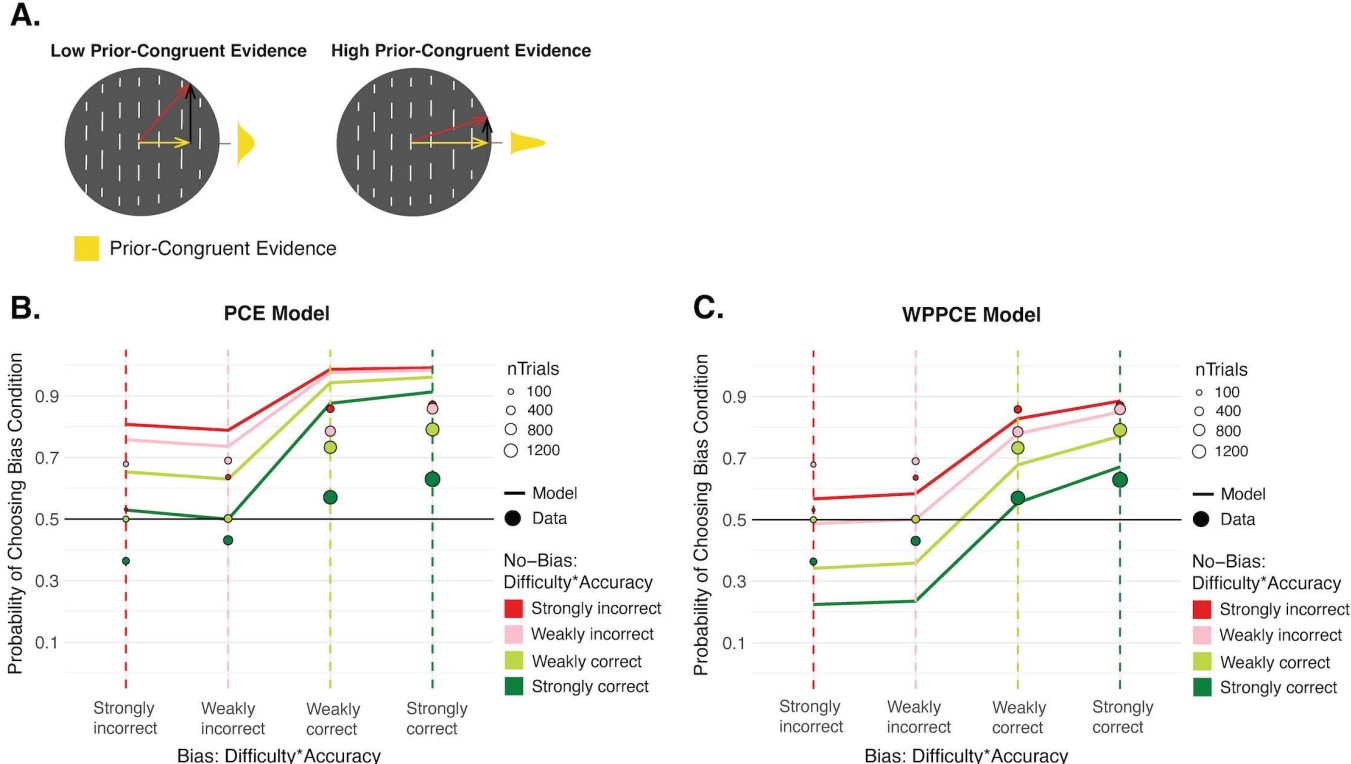

**Fig 6. Prior-Congruent Evidence models. A. Sketch of prior-congruent evidence.** In the line-motion stimuli, the evidence that is congruent with the prior is the component of the motion that is in the preferred direction, orthogonal to the lines' orientation, shown as the yellow vector component. The sensory evidence is shown in red and reflects the mean of the likelihood distribution, and the prior-incongruent component, parallel to the lines' orientation, is shown in black. Stimuli with high PCE have a motion direction that is closer to the preferred one, and therefore a longer vector component along that dimension. In the PCE confidence model, confidence was directly proportional to this PCE, and was not based on the posterior evidence or on the orange/blue decision made. **B. Confidence predictions from Prior-Congruent Evidence (PCE) model against data.** Confidence forced-choice predictions simulated from the PCE model (solid lines) against the confidence data (points). This model predicts more confidence choices favoring the Bias condition, in line with, though more extreme than, what is seen in the data. **C. Confidence predictions from Weighted Posterior and Prior-Congruent Evidence (WPPCE) model against data.** Confidence forced-choice predictions simulated from the WPPCE model with best-fitting α = 0.34 (solid lines) against the confidence data (points). In the WPPCE model, confidence is based on a weighted combination of two sources of evidence, the prior-congruent evidence shown in **(A)**, as well as the posterior evidence (as in the Bayesian model).

## Efficient Coding model

We also consider the possibility, in line with previous work [39–41], that there is efficient encoding of sensory evidence such that signals that are more likely to occur are encoded more precisely. This efficient encoding means that the likelihood precision itself is constrained by the prior, before the computation of the Bayesian posterior. So, this is another way in which the prior may effectively have a stronger impact than what is expected of the Bayesian model. Although efficient coding would dictate perceptual decisions as well, it impacts the shape of the posterior distributions, so it is possible that it can explain the observed dissociation between first-order decisions and confidence. We implemented an efficient encoding model following the approach of Wei and Stocker [39], which included a parameter capturing the variance of the prior, $\sigma_P$, (as in the Bayesian model), a parameter capturing neural noise, $\sigma_N$, and a parameter capturing transduction noise, $\sigma_T$, in the mapping from external stimulus to internal evidence. Transduction noise is necessary to account for variance in perceptual decisions and confidence values across trials, as otherwise the model would predict identical outcomes every time the same external stimuli are repeated. The model was fit to the perceptual decisions to give the best-fitting $\sigma_P$, $\sigma_N$,

and $\sigma_T$ parameters for each participant, and we then assessed its ability to capture the observed confidence patterns. While the predictions of this model better match the confidence data than the Bayesian model without efficient coding, it still underpredicts the confidence bias found favoring the Bias condition (Fig 7B). A more naive model that mirrored the Bayesian one but allowed the likelihood precision to differ between conditions also could not explain the observed bias at the confidence level (Fig G in S1 Text). So, while the results still agree with previous findings arguing that the brain may implement efficient coding, they also suggest that efficient coding cannot itself explain the dissociation found between the information used in perceptual decisions versus confidence.

## Model comparison

To compare the models quantitatively, we computed the AIC for each, therefore also penalizing the models with more free parameters. This revealed an $AIC_{Bayesian} = 30767.85$, $AIC_{Distance-to-Criterion} = 38774.61$, $AIC_{DP} = 29831.18$, $AIC_{PCE} = 31680.98$, $AIC_{WPPCE} = 25855.11$, and $AIC_{EC} = 25943.83$ (Fig 8A). The best model to explain the group confidence forced-choice data is the WPPCE confidence model (Fig 8B), in which confidence choices are based not only on the posterior evidence that formed the first-order decision, but additionally on the prior-congruent evidence as a separate cue. We additionally fit and compared the six models for each individual participant. This revealed the WPPCE confidence model to be the winning model in 15 out of the 24 participants, the EC confidence model to be the winning model in 7, and two participants for whom the model comparison was inconclusive (Fig 8C). Taken together, these results suggest that, relative to their first-order behavior, participants have a confidence bias that favors the condition in which the slow-motion prior more strongly influences decisions. Further, this bias can best be explained by a model in which participants use the prior-congruent information as an additional cue to inform their confidence, in combination with the Bayesian posterior evidence used for the motion-direction decisions.

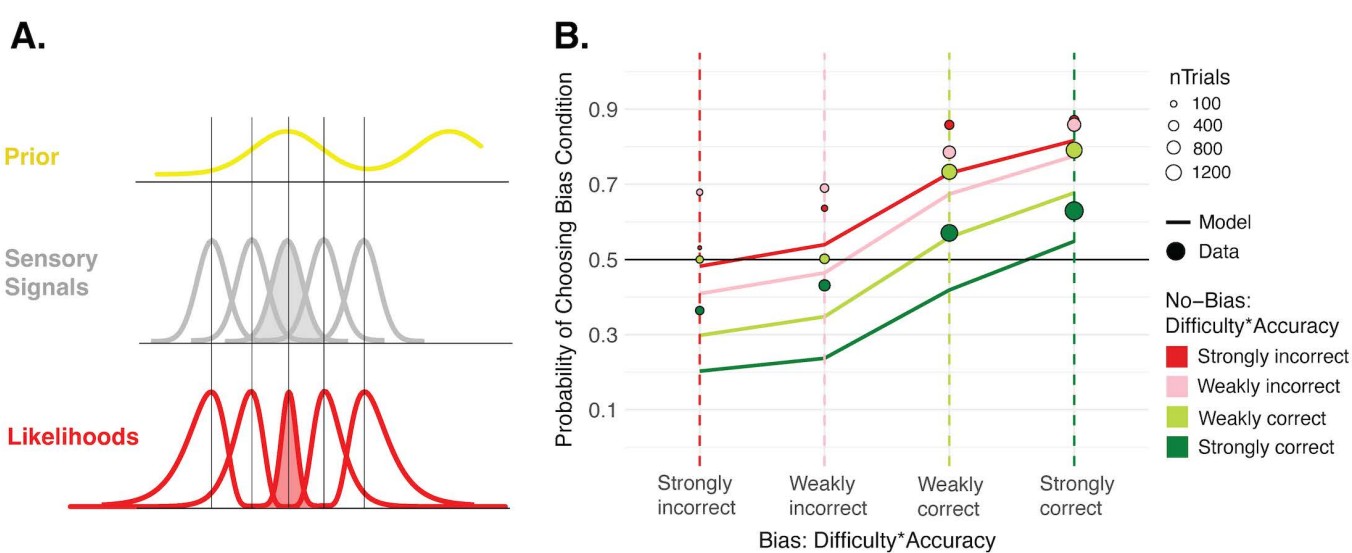

**Fig 7. Efficient Coding model. A. Sketch of efficient encoding.** The shape of the likelihood distribution on a given trial is constrained by how probable the stimulus signal is, based on the prior. Signals closer to the peak of the prior will be encoded more precisely, as indicated by the shaded red likelihood. The grey distributions reflect the encoding in a homogenous sensory space with uniform variance (neural noise). They map to the non-uniform likelihoods in physical space according to an inverse mapping described in Methods. **B. Confidence predictions from Efficient Coding (EC) model against data.** Confidence forced-choice predictions simulated from the EC model (solid lines) against the confidence data (points). Although this model better predicts confidence compared to the Bayesian model without efficient coding, it still cannot account for the observed bias at the confidence level.

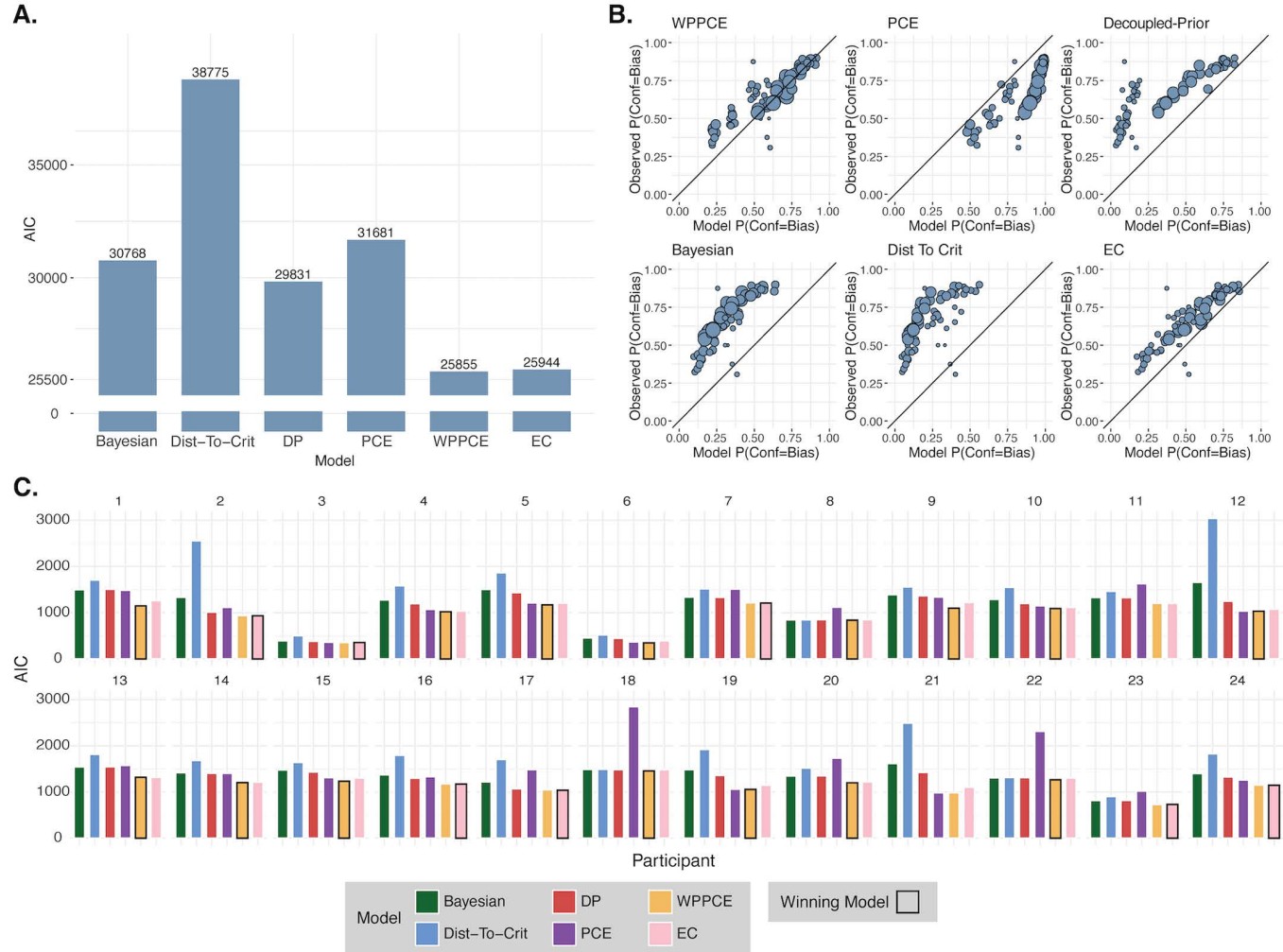

**Fig 8. Model comparison. A. AIC group results.** AIC comparison of models fit to the pooled group data. **B. Model goodness of fit.** Predicted against observed confidence choice rates from each of the models. Each point reflects a pair of stimulus settings with one interval in each condition and a given set of orange/blue responses, and the size of the point reflects the number of trials. The closer the points are to the x = y line, the better the model is at modeling confidence choice rates. Confidence choice rates reflect the probability of choosing the Bias condition as the more confident interval. So, points above the identity line reflect the model underestimating the frequency of confidence choices for the Bias condition, and points below the identity line reflect the model overestimating the frequency of confidence choices for the Bias condition. **C. Participant-wise AIC results.** AIC comparison of models fit to each individual participant. The winning model for each participant is shown outlined in black, when there was a conclusive winner (all except participants 11 and 21). The WPPCE model was the winning model for 15 participants and the EC model was the winning model for the remaining 7.

## Discussion

Previous work has revealed dissociations between the impact that priors have on confidence versus first-order decisions, suggesting a particularly strong role of priors in shaping confidence [18]. However, this previous work examined high-level priors that are induced flexibly in a task context, and may not be generalizable to perceptual priors that are formed naturally across the lifetime. Here, we investigated the role of priors in perceptual decisions and confidence using such a long-term perceptual prior, namely, the slow-motion prior. Using moving line stimuli, we created two conditions: one in which motion direction decision performance was strongly biased by the slow-motion prior (Bias condition), and another in

which no bias was induced and performance was driven by the stimulus information (No-Bias condition). We found that, even when any differences in first-order decision rates between conditions were accounted for, there was a confidence bias towards the Bias condition. This suggests confidence to favor the condition with more influence from the prior on decisions, rather than the condition with a more informative stimulus. By comparing computational models, we revealed this effect to be best explained by confidence choices using the prior-congruent evidence as an additional cue, over and above the posterior evidence that is used in perceptual decisions.

In line with findings using high-level, flexibly induced priors [18], the results here reveal that long-term perceptual priors also have a different influence on perceptual performance and confidence, with prior-congruent information having a particularly strong impact at the metacognitive level. This underlines a striking consistency of the observed confidence-performance dissociation across different types of priors. Because priors of these different categories are often argued to act and even be implemented differently [21,23], finding this dissociation to hold in the case of long-term priors is enlightening about its nature. Based on work using instructed probabilistic priors, it remained unclear whether their disproportionate impact on confidence was due to the abstract probabilistic information being very unlike (and therefore difficult to integrate with) perceptual information, or due to it acting at a late processing stage. However, long-term priors like the slow-motion prior are assumed to act directly at a perceptual level [26,27,32]. Therefore, the results here suggest that this pattern cannot be easily explained by high order, post-perceptual information acting to a greater extent on confidence. Another key difference between the high-level prior manipulations used previously and the low-level one used here is that participants were not aware of the influence of the slow-motion prior. This prior was naturally present in participants, not temporarily manipulated in the lab. Further, the stimuli were designed such that participants would not even notice a decision bias forming from the prior, since it biased their decisions evenly towards blue and orange response sectors. This rules out the possibility that the strong effect of priors on confidence can be explained by participants realizing that they had this prior information and incorporating it post-decisionally, or by a confidence bias induced by demand characteristics. The use of the confidence forced-choice paradigm further eliminates this possibility, reducing the chance of any report- or scale-based confidence biases [33].

Here we find confidence to be biased towards information that is consistent with one's prior beliefs. This relates to other confirmatory confidence biases in the literature, such as the positive evidence bias in which confidence is biased towards information that confirms one's decision [42,43]. Similarly to the positive evidence bias, we also find this to occur at a low level, and outside of the awareness of participants. Both of these confirmatory confidence biases may be adaptive for avoiding cognitive dissonance, and driving self-consistency [44,45]. Also, although confirming prior-congruent information may seem maladaptive for maintaining good predictive models, having a positive epistemic feeling that reinforces prior-congruent information might actually help drive us to minimize prediction errors by encouraging us to seek information that will be in agreement with expectations [1,46]. This might be particularly useful for a prior as stable and precise as the slow-motion prior. However, this form of confidence bias may in turn be limited to situations in which it is generally adaptive to confirm prior-congruent information or to be self-consistent. It is possible that, if the prior is unreliable, or when there is a decision bias that does not act as an informative prior at all, the metacognitive system might actually help us *discount* first-order bias, going in the opposite direction of our results. It would be interesting to extend the present research questions to such scenarios, for example by inducing false expectations, or by examining decision and confidence behaviour after change points of a volatile prior.

The findings here may also relate to another confidence-performance dissociation that has been discussed in the literature, in which confidence is influenced by the visibility of a stimulus, over and above the total decision evidence [47,48]. The weighted evidence and visibility (WEV) model captures confidence as a weighted combination of these two factors. In parallel with this, the prior-congruent evidence in the WPPCE model proposed here could reflect a visibility-like cue to confidence, if the prior-congruent information increases perceptual strength. This is in agreement with work finding that prior-congruent stimuli enter conscious awareness earlier, and at lower thresholds [49–54]. This would also predict

the trend we found towards a positive relationship between the strength of the perceptual prior and the degree of the PCE bias in confidence, though this should be investigated further since it did not reach significance. Beyond visibility, other perceptual features have also been suggested to have the potential to impact confidence, despite being decision-irrelevant [55]. Though the confidence forced-choice task has a benefit of reducing confounding confidence biases, it also limits our insight into the subjective quality of the effect. Future work could explore in more depth how information in the preferred orthogonal motion direction impacts the phenomenal experience of the stimuli, which may clarify the relationship between the findings here and other confidence biases.

This work is also pertinent to an ongoing discussion in the literature about whether confidence computations are Bayesian [5,6,8,56]. Assessments of the Bayesian confidence hypothesis have primarily focussed on testing whether confidence computations are probabilistic in nature and capture the probability of a decision being correct, rather than using simpler heuristics [57,58]. However, this literature has largely ignored a central aspect of the broader Bayesian framework: the role of priors. For confidence to be Bayesian, it is critical that it tracks the *posterior* probabilities, incorporating not just sensory evidence but also the prior. We focussed on understanding this aspect of confidence. Further, we specifically examined low-level, long-term priors, which are understudied in the confidence literature. We revealed that confidence *does* incorporate information from a long-term prior of this type, as would be expected of Bayesian confidence. In one sense, this provides new support to the Bayesian confidence hypothesis, as it demonstrates the computation of the posterior probability correct under informative priors. At the same time, we found deviations from the Bayesian model, with the use of an additional cue that biased confidence. Overall, we do not take these findings as strict evidence either for or against the Bayesian confidence hypothesis, as we did not design the experiment to test it, but rather take the approach – argued for in the literature [59] – of using the Bayesian model as a normative benchmark from which to better understand the information impacting confidence.

## Limitations

Although this work extends previous findings to what is often considered a different category of prior, it is still only one example of a long-term prior, and it is possible that others act in different ways. In line with this possibility, recent work examining confidence under the influence of natural image statistics found different effects of an orientation prior versus a lighting prior on confidence [19]. It would therefore be valuable to find ways to investigate other examples. Also, although we see a similar confidence-performance dissociation with a long-term prior as was previously found with high-level priors, we cannot draw any conclusions about a shared mechanism of action. Future studies testing for correlations between the strengths of these dissociations across different priors, as well as neuroimaging work exploring this effect in terms of neural implementation would be useful to clarify this.

While the conditions we created and models we tested allowed us to quantitatively compare prior-use in decisions and confidence, they also came with some limitations. Our use of a binary discrimination task for the perceptual judgments allowed us to build the conditions such that the prior only influenced decision rates in the Bias condition. However, there could still be residual effects of the prior on perception in the No-Bias condition that do not impact performance, which could be explored in future work using a reproduction task. It is also possible that participants formed other short-term priors about the task structure. We tried to avoid this by having the line orientations and absolute motion directions sampled nearly uniformly across the full 360° range, making conditions and stimulus settings difficult to detect. But, future work could instead use a more conspicuous task structure and then model participants' learning of it. This would make a more complete Bayesian Optimal Observer model, whereas here in our Bayesian model we focus specifically on a Bayesian use of the slow-motion prior in decisions and confidence. We also note that here we chose to model the prior in terms of the directional bias that is created in these stimuli, rather than modeling the prior over speed. This was for computational tractability of the confidence models as well as simplicity of interpretation, since the directional prior reflects the focus of the motion direction task better. Further, the effect of the slow speed prior on the perceived motion direction bias has been

explored extensively in previous work [26,27,32,60] and was not central to our research questions, which only required that we could well capture the resulting bias effect in decisions and confidence. Still, it could be interesting for future work that is more focussed on understanding the nature of the slow speed prior to explore these questions in a speed estimation task. Finally, though the WPPCE model could best explain the data, and predicts the confidence bias of interest, it still does not perfectly capture the observed confidence patterns in their entirety. This may be in part due to noise in the data, particularly since the confidence forced-choice paradigm was complex for participants and we could not force balanced trial numbers across all combinations of stimuli and responses, so some settings were very sparsely sampled. However, it would be valuable to continue exploring extensions to the model that may allow it to better predict confidence choices, such as quantifying metacognitive noise as an additional parameter.

## Conclusion

While there is evidence from previous work that priors impact first-order decisions and confidence in different ways, it had previously remained unclear whether this would generalize to long-term perceptual priors. Here, we found a similar dissociation in the effect of the slow-motion prior on first- versus second-order processing, with the prior-congruent evidence more strongly impacting confidence. This suggests a confirmatory confidence bias favoring evidence congruent with priors, which occurs implicitly and from low-level priors that are naturally formed in participants. It is therefore important to account for this effect in order to model and understand confidence across the variety of naturalistic situations in which priors influence our processing.

## Materials and methods

The experiment was pre-registered (https://osf.io/3uc4k), and we respected the pre-registered plan unless stated otherwise.

### Ethics statement

Participants were compensated with 10€ per hour and gave signed, informed consent before starting the experiment. The local ethics committee (Comité d'Ethique pour les Recherches en Santé) approved the study, which conformed to the Declaration of Helsinki.

### Participants

We pre-registered that we would test 25 participants, ensuring at least 20 clean datasets that met the prespecified inclusion criteria, which most importantly included that they showed the basic effects of the prior – that the perceptual decision rates were biased towards the orthogonal motion direction in the Bias condition. We tested 25 participants and later excluded the data from one of them because they did not show the basic effect of the prior, leaving data from 24 participants (7 male, 17 female) included in our analyses. Participants were between 21 and 39 years of age (M=24.76, SD=4.28), reported to have normal or corrected-to-normal vision and were fluent in English.

### Setup

The experiment was programmed using MATLAB [61] and Psychtoolbox-3 [62–64]. Participants sat in a dark room with their head on a chin and forehead rest placed 57 cm away from a CRT monitor (Vision Mater Pro 454) with a display resolution of 1,920 x 1,200 (refresh rate=60Hz).

### Procedure

**ASA procedure.** At the beginning of the first session, after receiving verbal instruction about the stimuli, participants completed a short stimulus training consisting of 80 trials across five mini-blocks (16 trials each) in which the line motion stimulus was shown and they input their answer about whether the lines were moving towards the orange or blue region

using the mouse. For the first mini-block, the true line motion direction was displayed. In addition, the lines started at high contrast so that participants could clearly see and understand the stimuli, and got lower contrast in later blocks (Weber contrasts: 22%, 22%, 6%, 6%, 6%), and their movement was displayed for a longer duration at first and got progressively shorter in each block (durations: 1000ms, 266ms, 133ms, 133ms, 133ms). After this training, participants completed a staircasing procedure, which took approximately 10 minutes. We used this to select the appropriate θ values for each individual that would target the desired perceptual decision rates in each condition, given their performance and the strength of their bias. We ran an accelerated stochastic approximation (ASA) staircasing procedure [65] once in the No-Bias condition, with the reference point exactly 90° away from both preferred, orthogonal line orientation directions. For the Bias condition, we ran two staircasing procedures, to capture the bias effect from the influence of the prior in both directions (a blue-bias and an orange-bias), by setting the preferred, orthogonal line orientation either 35° towards the blue region or towards the orange region from the reference (Fig 1A). For each of these staircasing procedures, four staircases controlled the motion direction (relative to the reference) and were set to converge to two decision probability thresholds of 0.25 and 0.75. The staircasing procedures were all interleaved, meaning that the conditions were also interleaved. We then fit a cumulative normal distribution function (Φ) with a free noise ($\sigma_{asa}$) and bias ($\mu_{asa}$) parameter to the data from each of these three separate ASA procedures:

$$P = \Phi(\theta; \mu_{asa}, \sigma_{asa})$$

where $P$ is the decision probability and θ is the stimulus value used, which was the angle between the reference and the motion direction (Fig 1C). There was no lapse rate parameter to secure robust fitting during the experiment. These fit psychometric functions were then used to select the stimulus values that targeted probabilities of choosing blue (P('Blue')) of 0.15 and 0.35 when there was an orange bias, and 0.65 and 0.85 when there was a blue bias in the Bias condition, and probabilities of choosing blue of 0.15, 0.35, 0.65, and 0.85 in the No-Bias condition. These stimulus values were used for the main task. We did not include all four decision probabilities with both bias directions (blue and orange) because we did not want to examine cases in which the bias worked against the stimulus and hence the Bias condition actually required *more* stimulus information than the No-Bias condition. Including those trials would have substantially increased the length of the experiment, and would have caused an imbalance in the number of trials across conditions, without adding much to our analyses. Such extreme Bias condition cases (in terms of θ) may also have risked the participants noticing the two conditions.

**Main task.** After completing the ASA procedure, participants received verbal instructions about the structure of the confidence forced-choice task, and completed another short training of 8 trials. Each trial of the main task consisted of two consecutive motion direction decisions followed by a confidence choice. We used this criterion-free confidence forced-choice structure in order to reduce confidence biases that come from criterion placement or particular use of a confidence scale, which can make the interpretation of confidence results difficult [33]. The motion direction decisions worked in the same way as in the ASA procedure – participants viewed the set of moving lines at short duration (133ms) and low contrast (7.2% Weber contrast) and then used the mouse to select whether the motion direction was towards the orange area or blue area by clicking on the corresponding colored quarter-ring around the display circle. In the confidence choice, participants used the left and right arrow keys to indicate whether they were more confident in the first or second interval, respectively (Fig 1B). We used the chosen stimulus values (θ's, four per condition) from the ASA procedure to create 28 possible pairs, by using all possible pairings of these eight stimulus values except for ones with the identical stimulus value and condition twice. These 28 pairs formed the possible interval pairs and were counterbalanced across the main task in both possible interval orders. We fixed each block such that half the trials would have the line orientation rotated clockwise from vertical, and half counterclockwise from vertical, and amount of rotation was sampled randomly on each trial from the possible values: 10°, 20°, 30°, 40°, 50°, 60°, 70°, or 80°. We sampled in this way and excluded the

cardinal orientations (0° and 90°) in order to avoid any overall effects of the cardinal-direction prior, a different long-term prior that biases processing of orientations towards the cardinal axes [66]. We also fixed each block such that half the trials would have orange and half would have blue as the ring color counterclockwise to the reference. After assigning the line orientation and the placement of the blue and orange regions, the rotation of the reference and then line motion direction were set relative to these, and were dictated by the stimulus value pair – the condition dictated the placement of the reference and the θ dictated the line motion direction (Fig 1A). If the stimulus setting dictated a true line motion direction towards blue, the θ would be set so that the lines moved towards the region colored blue on that trial. In the Bias condition, if the stimulus setting dictated a 'Blue Bias', the reference would be set such that the preferred motion direction fell in the region colored blue on that trial. The order of trials was randomized within each block.

Participants completed 24 repetitions of each of the 28 stimulus pairs for a total of 672 trials, including 672 motion direction decisions in each condition (for a total of 1344 motion direction decisions) and 672 confidence choices. These took place across 12 blocks with 56 trials in each block. Four of these blocks were completed in the first session and the remaining eight took place in the second session, with the sessions occurring either on the same day with a minimum of one hour break between them, or on consecutive days. With the time taken for instructions and the ASA procedure in the first session, this resulted in each session taking approximately 1 hour.

**Stimuli.** The line motion stimulus was similar to the stimulus used in work by Sotiropoulos et al. [32]. It consisted of a matrix of parallel line segments that moved rigidly, all in the same direction, and at a constant velocity. The line movement speed was 3° of visual angle per second and they were shown for a duration of 133ms. The matrix of lines was displayed through a circular mask in the center of the screen, which was 12° of visual angle in diameter. The line segments making up the matrix were 2.86° of visual angle in length, and were hence much shorter than the circular mask, allowing many endpoints to be visible. This is critical because only the line ends provide the disambiguating information regarding the line velocity. The lines were 3.0 arcmin thick. The background was middle gray with a luminance of 0.5 cd/m$^2$ and the lines were lighter gray with a luminance of 0.536 cd/m$^2$, giving a Weber contrast of 7.2%. For the orange/blue motion direction decision, the orange and blue decision regions were displayed by showing a colored half-ring around 180° of the circular mask, with 90° of that being blue and 90° of it being orange. This half-ring was visible for the duration of the line motion in order to avoid it appearing after and masking the stimulus. Participants then reported their decision by clicking the desired region of this half-ring. When participants moved the mouse close enough to one of the decision regions to select it, the orange or blue segment expanded slightly to highlight it, such that participants knew which they were selecting. This was helpful because the mouse cursor was invisible in order to avoid distracting participants during the stimulus display, or having a masking effect if it appeared immediately after. Because of this, the cursor always started in the center, which participants were instructed about, and they had a chance to adjust to this in the stimulus demo. Before each stimulus, a green fixation point in the center of the screen was displayed for 500ms. The half-ring only appeared after this, with the matrix of lines, in order to avoid participants planning a strategy or fixating on the reference prior to the stimulus onset.

## Analysis

As pre-registered, we removed any trials with reaction times that were longer than 8 seconds for any decision, or shorter than 100ms for any motion direction decision. It was also possible that some participants failed to demonstrate the expected perceptual bias, at least for some stimulus settings (such as for a particular direction of bias, or at some difficulty levels), due to individual differences in the strength of the prior. So, we also removed the data from any stimulus settings for any given participant in which the basic bias effect was not present. This was determined to be the case if, as pre-registered, the selected θ value (from the ASA procedure) targeting a given perceptual decision rate in the No-Bias condition was not more extreme than the θ value targeting the same perceptual decision rate in the Bias condition. Trials with the same condition in both intervals were also primarily included to constrain the fitting of the cfc-model,

and were removed from several of the other analyses, such as the non-parametric confidence analysis and the modeling building on the Bayesian model.

Our behavioral analysis testing the effect of condition on motion direction decisions was done using a logistic mixed-effects model with the 'lme4' package [67] in R [68]. The test was two-tailed and used an alpha value of 0.05. We did not assess confidence choices using a regression approach because we prespecified that we would only do this if the first-order decision probabilities were adequately matched, which they were not. Additionally, we realized from further model exploration that the Bayesian model does not predict equal confidence given matched first-order decision rates, so this analysis was not the best approach to assess deviations from the Bayesian benchmark in any case. Instead, and as planned given that first-order decision rates may not be perfectly matched, to examine confidence bias we used a non-parametric analysis following a similar approach to that in previous work [35]. We transformed the perceptual decision probabilities to put them in the $(-\infty,\infty)$ domain by taking their log-odds, and then computed the differences between the two conditions for each stimulus pair setting, excluding pairs with the same condition in both intervals. For each of these transformed choice probability differences ($cp_{diff}$), we then computed the corresponding rate of choosing the Bias condition as the more confident one ($P(\text{"Bias"})$) and fit a cumulative normal distribution function ($\Phi$) with a free noise ($\sigma_{cb}$) and bias ($\mu_{cb}$) parameter to this relationship:

$$P(\text{"Bias"}) = \Phi\left(cp_{diff}; \mu_{cb}, \sigma_{cb}\right)$$

## Modeling

**Quantifying confidence bias.** In order to further quantify the confidence bias, before exploring the different process models that might explain it, we fit the *cfc-model* developed in previous work [33]. This model is based on elements of signal detection theory [69] and includes two main parameters. The confidence noise parameter accounts for inefficiencies in using all the sensory information for the confidence judgment, and the confidence boost parameter reflects the extent to which confidence is based on the same information as that used to commit on the perceptual decision or new sensory information. For our purposes, we focus on a third parameter, the confidence bias term, $\beta$, which captures the propensity to choose one condition as more confident when it is competing with another condition, for instance, central versus peripheral vision [35]. Importantly, this parameter is computed after accounting for the stimulus strengths, first-order responses in each interval, and a potential interval bias (propensity to prefer a particular interval as the more confident one). In a single condition, confidence bias would reflect the extent to which the sensory sensitivity is correctly estimated, such that if this parameter is equal to 1, sensitivity is perfectly estimated. If this parameter is below 1, sensitivity is underestimated indicating underconfidence and if this parameter is above 1, sensitivity is overestimated indicating overconfidence. While we cannot estimate confidence bias in a single condition (the bias would affect equally both intervals and thus cancel out any effect), we can compare confidence biases across two conditions, as in our case. Here, the *cfc-model* fits $\beta$ as the ratio of confidence biases between conditions with the No-Bias as the baseline, such that $\beta = 1$ indicates no confidence bias. If $\beta > 1$, this would indicate a confidence bias favoring the Bias condition, and $\beta < 1$ would indicate a confidence bias favoring the No-Bias condition. In order to collapse the Bias condition into one psychometric function for the *cfc-model*, we projected the orange bias trials onto the blue bias space by taking:

$$\theta_{Orange \to Blue} = \mu_{No-Bias} - \theta_{Orange}$$

and

$$Response_{Orange \to Blue} = 1 - Response_{Orange}$$

We fit the model to the pooled data, using normalized stimulus intensity values ($\theta_{norm}$) that were adjusted based on each participants' fit sensory noise ($\sigma_{No-Bias}$) and bias ($\mu_{No-Bias}$) from the No-Bias condition:

$$\theta_{norm} = \frac{\theta - \mu_{No-Bias}}{\sigma_{No-Bias}}$$

The model freely fit the sensory noise and criteria in each condition, along with the confidence parameters: confidence noise in each condition, confidence boost (reflecting additional information used in confidence) in each condition, interval bias, and the confidence bias parameter, $\beta$. We did this for 100 bootstrapped runs of the model in order to extract the confidence intervals surrounding the estimate of $\beta$. The other fitted confidence parameter values are reported in Supporting Information (Table A in S1 Text). We implemented and fit this model using the previously developed 'cfc' software [70]. More on the details underlying the model and the other parameters fit can be found in the original work [33].

**Bayesian decision model.** For the remaining modeling, we wanted to assess whether the information used in confidence computations deviates from that used in perceptual decisions, and in particular whether the use of the slow-motion prior deviates. Hence, we took the approach of fitting a model to the first-order perceptual decision data, and then exploring whether deviations from that model are needed to explain confidence. So, the first-order parameters were fit exclusively to the perceptual decisions, and any further confidence parameters ($w$ in the Decoupled Prior model and $\alpha$ in the WPPCE model) were fit exclusively to the confidence choice rates (per set of perceptual responses). To capture the first-order motion direction decisions, we built a Bayesian observer model (Fig 3A). It has a Gaussian mixture prior distribution of motion directions (in degrees away from the reference) with means at the two preferred, orthogonal motion directions ($\mu_{orth1}$, $\mu_{orth2}$),

$$prior \sim [0.5 * N(\mu_{orth1}, \sigma_P) + 0.5 * N(\mu_{orth2}, \sigma_P)]$$

where $\sigma_P$ is the standard deviation of each Gaussian making up the mixture prior. Due to low $\sigma_P$ values and a decision region restricted to between -90° and 90°, wrapping was not an issue for these Gaussians. In the No-Bias condition, $\mu_{orth1}$ and $\mu_{orth2}$ are at + and -90°. In the Bias condition, $\mu_{orth1}$ is 35° away from the reference in the bias direction, and $\mu_{orth2}$ is 145° away from the reference in the non-bias direction. The Gaussian likelihood is formed from the stimulus corrupted by sensory noise,

$$likelihood \sim N(r, \sigma_L)$$

where $\sigma_L$ is the sensory noise and $r$ is the internal signal caused by the stimulus. These combine to form the posterior distribution, and the perceived probability of the motion direction being in the orange (encoded as the negative direction) or blue (encoded as the positive direction) region is computed by the area under this posterior distribution across the entire orange or blue region, normalized:

$$P(Orange) = \frac{1}{N} \int_{-90}^{0} [0.5 * \varphi(x; \mu_{orth1}, \sigma_P) + 0.5 * \varphi(x; \mu_{orth2}, \sigma_P)] * \varphi(x; r, \sigma_L) \, dx \tag{1}$$

$$P(Blue) = \frac{1}{N} \int_{0}^{90} [0.5 * \varphi(x; \mu_{orth1}, \sigma_P) + 0.5 * \varphi(x; \mu_{orth2}, \sigma_P)] * \varphi(x; r, \sigma_L) \, dx \tag{2}$$

where $N$ is equal to the total area under the posterior in the allowed decision region:

$$N = \int_{-90}^{90} [0.5 * \varphi(x; \mu_{orth1}, \sigma_P) + 0.5 * \varphi(x; \mu_{orth2}, \sigma_P)] * \varphi(x; r, \sigma_L) \, dx. \tag{3}$$

The decision is then based on the max of these two values,

$$decision = max\left[P(Orange), P(Blue)\right].$$

We fit this model to the motion-direction decisions of each individual participant in order to find the $\sigma_P$ and $\sigma_L$ values that would best explain their first-order performance. However, in order to deal with the computational demand of this we required a slight simplification. For the Bias condition, we considered the prior to be a single Gaussian centered around $\mu_{orth1}$. Because the Gaussian centered around $\mu_{orth2}$ (+-145°) is so far outside of the allowed decision region (from -90° to +90°), this had a negligible impact on the model and made it possible to fit with the amount of data available to us. The ability of the model to still account well for first-order performance can be seen in Fig 3C. We then used the fit $\sigma_P$ and $\sigma_L$ parameters for the rest of the confidence modeling described below, with the exception of the Efficient Coding model, which fit a different perceptual decision model as a baseline, based on the idea of efficient coding.

**Bayesian confidence model.** We then considered the expected confidence patterns of an observer that follows the Bayesian confidence model. This model is in line with the Bayesian confidence model discussed in the literature in which confidence reflects the perceived posterior probability of being correct about a decision [5–10]. In our case, confidence then corresponds to the perceived posterior probability that the orange/blue decision is correct, which is described above in Equation (1) if orange is chosen, and [2] if blue is chosen. The confidence forced-choice after both intervals is then based on whichever of these two confidence values from the two intervals is higher:

$$cfc = max[conf1, conf2],\tag{4}$$

where the *conf* value is the perceived posterior probability correct. This model has no free parameters beyond the first-order sensitivity parameters, $\sigma_P$ and $\sigma_L$.

**Distance-to-Criterion confidence model.** It is possible that, instead of computing the area under the posterior distribution as in the Bayesian confidence model, participants instead just base their confidence on one sample from the posterior. In this model, confidence reflects the distance between a sample from the posterior ($r_c$) and their criterion (*crit*):

$$conf = \left| r_c - crit \right|\tag{5}$$

High confidence decisions then occur when the internal posterior sample is far from the criterion, or is more clearly orange or blue. Again, the confidence forced-choice is then based on which of these confidence values is higher, as in Equation (4). We also assume in this model that the sample on which confidence is based is on the same side of the criterion as the decision, such that confidence will not oppose the decision. Like the Bayesian model, this model has no added free parameters for confidence.

**Decoupled Prior confidence model.** The Decoupled Prior confidence model reflects participants computing confidence as in the Bayesian confidence model above, but allowing for a different estimated precision of the prior relative to the likelihood such that the prior may have a stronger or weaker effect in confidence compared to the first-order decisions. In this sense, the prior used in confidence decouples from that used in perceptual decisions. The relative over- or underweighting of the prior is implemented by an over- or underestimation of the sensory noise of the likelihood through a scalar weighting parameter *w*, such that the estimated posterior probabilities of each choice at the confidence level are:

$$P(Orange) = \frac{1}{N}\int_{-90}^{0}\left[0.5 * \varphi\left(x; \mu_{orth1}, \sigma_P\right) + 0.5 * \varphi\left(x; \mu_{orth2}, \sigma_P\right)\right] * \varphi\left(x; r, w * \sigma_L\right)dx\tag{6}$$

$$P(Blue) = \frac{1}{N}\int_{0}^{90}\left[0.5 * \varphi\left(x; \mu_{orth1}, \sigma_P\right) + 0.5 * \varphi\left(x; \mu_{orth2}, \sigma_P\right)\right] * \varphi\left(x; r, w * \sigma_L\right)dx\tag{7}$$

and *N* is:

$$N = \int_{-90}^{90} [0.5 * \varphi(x; \mu_{orth1}, \sigma_P) + 0.5 * \varphi(x; \mu_{orth2}, \sigma_P)] * \varphi(x; r, w * \sigma_L) \, dx \tag{8}$$

If $w = 1$, the sensory precision is correctly estimated in the confidence computation and the prior and likelihood are precision-weighted in the same way as in the first-order decisions. This, additionally, would be identical to the Bayesian confidence model. If $w < 1$, the sensory noise is underestimated, and the prior is underused relative to the likelihood. On the contrary, if $w > 1$, the sensory noise is overestimated, and the prior is overused relative to the likelihood. In this model, the scaling parameter $w$ was fit as a free parameter. Confidence was allowed to vary between 0 and 1, so there could be cases in which, due to the different prior-use at the confidence level, it was below 0.5 and hence did not agree with the perceptual decision. Such cases reflect the possibility of a change of mind following the perceptual decision, with confidence in the originally chosen option then being particularly low.

**Prior-Congruent Evidence (PCE) confidence model.** In the PCE confidence model, confidence is not based on the Bayesian posterior distribution but is instead based on the amount of prior-congruent information in the stimulus. For these line motion stimuli, the prior-congruent information refers to the information in the preferred, orthogonal motion direction (Fig 6A). So, the confidence is proportional to the length of the vector component in this orthogonal direction,

$$conf = PCE = \left| cos(\theta_r - \theta_P) \right| \tag{9}$$

where $\theta_r$ is the angle between the reference and the motion direction of the internal signal generated by the stimulus, and $\theta_P$ is the angle between the reference and the preferred, orthogonal direction. The confidence choice is then based on which of these *PCE* values is larger, or which of the two intervals had more prior-congruent evidence, independently of the posterior evidence and the first-order responses.

**Weighted Posterior and Prior-Congruent Evidence (WPPCE) confidence model.** The WPPCE confidence model suggests that the prior-congruent evidence described above serves as an additional signal, but not the sole signal, to confidence. In this model, confidence is based on the weighted combination of this *PCE* and the Bayesian posterior evidence (*BPE*), weighted according to a weighting parameter α,

$$conf = (1 - \alpha) * BPE + \alpha * PCE \tag{10}$$

where the *PCE* term is computed as in Equation (9), and the *BPE* term is computed as in Equation (1) or (2), depending on the perceptual decision. Larger values of α indicate more use of the prior-congruent evidence in confidence, and smaller values of α indicate more use of the posterior evidence. When $\alpha = 0$, this model is identical to the Bayesian confidence model and when $\alpha = 1$, this model is identical to the PCE confidence model. The weighting term, α, was a free parameter.

**Efficient Coding (EC) model.** The efficient coding idea suggests that the brain should allocate more resources to represent stimuli that are more likely to occur. The implementation of efficient coding in the model here follows that of Wei and Stocker [39,71]. They formalize the link between allocated resources – as measured by the Fisher information of the sensory evidence, $J(\theta)$, – and the stimulus probability – the prior, $p(\theta)$, – such that $p(\theta) \propto \sqrt{J(\theta)}$. A function $F(\theta)$, which is defined as the cumulative of the prior, maps the physical stimulus ($\theta$) to a sensory space with uniform Fisher information, yielding $\tilde{\theta}$. We then fully define the likelihood distribution in this sensory space by assuming Gaussian neural noise of standard deviation $\sigma_N$, and a mean that is jittered from $\tilde{\theta}$ due to transduction noise, $\sigma_T$. The likelihood in physical space is obtained using the inverse mapping $F^{-1}(\tilde{\theta})$. The resulting likelihood will not be homogeneous in physical stimulus space and will often be asymmetrical (Fig 7A). All other computations are identical to the Bayesian model (Equations 1–4), but using this differently defined likelihood. This means that the likelihood, which is already constrained by the prior, then

gets combined with the prior in the Bayesian integration to form the posterior, effectively giving an *additional* impact of the prior on processing. As with the Bayesian model, the free parameters were fit to the perceptual decisions and we then simulated the confidence choices that would be made of an observer using the same (efficiently encoded) information in their confidence as in their perceptual decisions. The free parameters here were the variance of each Gaussian comprising the prior, $\sigma_P$, the neural noise $\sigma_N$, and the transduction noise $\sigma_T$.

**Model-fitting and comparison.** The confidence models with additional free parameters for confidence – the Decoupled Prior model and the WPPCE model – were fit using a maximum likelihood estimation (MLE) approach to find the $w$ and $\alpha$ parameter values respectively that could best explain the confidence forced-choice data. Due to model complexity, we fit the three free parameters of the EC model to the perceptual decisions of each participant using a grid-search across a parameter grid spanning all combinations of $\sigma_P$ from 20 to 100 in steps of 5, $\sigma_N$ from 0.02 to 0.5 in steps of 0.02, and $\sigma_T$ from 0.02 to 0.1 in steps of 0.02, to find the maximal likelihood. We used a simulation-based approach to approximate the likelihoods for the EC, Decoupled Prior, and WPPCE models, which were analytically intractable. This was also done to compute the likelihood of the other models, to assess model fit. Model fitting was done in R using the 'stats4' package. In fitting the Decoupled Prior model, we also did an initial grid search across values of $w$ from 0.1 to 4, in steps of 0.1, and used the $w$ value that corresponded with the highest likelihood as the start value of the MLE procedure. This was done to avoid convergence issues. For a model comparison, we compared AIC between the models, computed according to

$$AIC = 2 * k - 2 * ln(\hat{L})$$

where $k$ is the number of free parameters and $\hat{L}$ is the maximized value of the likelihood function. We considered a model to be conclusively better if its AIC was lower by at least 2, following convention in the literature. For completeness, we also compared the models in terms of BIC, which did not change the results (Fig H in S1 Text). We ensured the models to be distinguishable from one another in a model recovery analysis (Fig I in S1 Text).

## Supporting information

**S1 Text. Supporting information.** Manipulation check per participant; Psychometric functions fit to motion direction decisions per participant; Non-parametric analysis of confidence bias per participant; Confidence choice results split by stimulus and perceptual decision; Posterior Mean model; Correlation between perceptual bias and confidence bias; Condition-specific Likelihood model; CFC-model parameters; BIC results; Model recovery analysis.
(DOCX)

## Acknowledgments

Part of this work was presented at VSS 2024.

## Author contributions

**Conceptualization:** Marika Constant, Elisa Filevich, Pascal Mamassian.

**Data curation:** Marika Constant.

**Formal analysis:** Marika Constant.

**Funding acquisition:** Elisa Filevich, Pascal Mamassian.

**Investigation:** Marika Constant.

**Methodology:** Marika Constant, Pascal Mamassian.

**Supervision:** Elisa Filevich, Pascal Mamassian.

**Visualization:** Marika Constant.

**Writing – original draft:** Marika Constant.

**Writing – review & editing:** Marika Constant, Elisa Filevich, Pascal Mamassian.

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
