## [Decision Letter · Decision Letter 0]

30 Apr 2025

PCOMPBIOL-D-25-00503

Long-term perceptual priors drive confidence bias that favors prior-congruent evidence

PLOS Computational Biology

Dear Dr. Constant,

Thank you for submitting your manuscript to PLOS Computational Biology. After careful consideration, we feel that it has merit but does not fully meet PLOS Computational Biology's publication criteria as it currently stands. Therefore, we invite you to submit a revised version of the manuscript that addresses the points raised during the review process.

Please submit your revised manuscript within 60 days Jun 30 2025 11:59PM. If you will need more time than this to complete your revisions, please reply to this message or contact the journal office at ploscompbiol@plos.org. Please include the following items when submitting your revised manuscript:

We look forward to receiving your revised manuscript.

Kind regards,

Wolfgang Einhäuser

Academic Editor

PLOS Computational Biology

Lyle Graham

Section Editor

PLOS Computational Biology

**Additional Editor Comments :**

As you will see, the reviews are positive in general, but the reviewers also raise plenty of issues, in particular regarding the modelling, that need to be carefully addressed.

**Journal Requirements:**

2) Your manuscript is missing the following section heading: Introduction. Please ensure all required sections are present and in the correct order. Make sure section heading levels are clearly indicated in the manuscript text, and limit sub-sections to 3 heading levels. An outline of the required sections can be consulted in our submission guidelines here:

4) We notice that your supplementary Figures, and Table are included in the manuscript file. Please remove them and upload them with the file type 'Supporting Information'. Please ensure that each Supporting Information file has a legend listed in the manuscript after the references list.

Potential Copyright Issues:

i) Figure 1. Please confirm whether you drew the images / clip-art within the figure panels by hand. If you did not draw the images, please provide (a) a link to the source of the images or icons and their license / terms of use; or (b) written permission from the copyright holder to publish the images or icons under our CC BY 4.0 license. Alternatively, you may replace the images with open source alternatives. See these open source resources you may use to replace images / clip-art:

**Reviewers' comments:**

Reviewer's Responses to Questions

Reviewer #1: Review of "Long-term perceptual priors drive confidence bias that favors prior-congruent evidence"

Reviewer: Michael Landy

This is a nice paper that asks whether built-in long-term priors, which bias first-order estimates of stimulus properties, also have an effect on confidence judgments about first-order judgments. They cleverly do this with a prior that acts somewhat implicitly by using the slow-speed prior, which in addition to biasing speed estimate (e.g., for low-contrast stimuli), also biases motion-direction estimates from linear stimuli. Using a stimulus consisting of a set of short line segments in an aperture, they independently vary the orientation of the line segments and the motion direction of the line texture. When the line orientations are parallel to the motion direction, the tendency to see motion orthogonal to these contours is irrelevant, leading to no bias in motion-direction judgments. On the other hand, when the line orientations are close to orthogonal to the motion direction, perceived direction is biased toward the nearest orthogonal orientation of the line texture. The question is how will this bias affect confidence in judgments of motion direction. They do this using the lab's confidence forced-choice task and all possible (unequal) pairs drawn from 8 stimuli (4 are biased and 4 unbiased). They also propose and compare a variety of models of the behavior and find that judgments are best accounted for by a model that combines a Bayesian confidence with a second component that favors prior-contingent evidence.

All in all, I think this is an interesting and a strong paper. However, given my foggy memory, I just read it straight through and found that the exposition was needlessly confusing and could be made much easier on the reader in spots. The main thing I tripped on a bunch was what the conditions were, what the pairs were, and which pairs went into which analysis. Eventually, all of this is described somewhere, but if you read it from the beginning, it's easy to get confused. Why? Well for one thing, in several places it is stated that there are four bias-condition stimuli and four no-bias stimuli. But, there are also 4 levels of matched choice probabilities (.15, .35, .65, .85) and two kinds of biases (orange vs. blue). Thus, one might easily conclude that there must be 12 kinds of stimuli, not 8, and that maybe they are pooling some of them somehow. I only got un-confused when I save Figure 2A in which we find out that only two choice-probabilities are used for each of the two colors. Why? It's never said. I still don't know why you didn't include the other obvious two points on the blue and orange curves. Maybe, as in Locke, Gaffin-Cahn et al. (2020), those extra two conditions would lead to such biased choices that they wouldn't add any leverage to the modeling and would just increase the number of pairs (from 28 to 66) too much. It would be helpful to clarify this early and to motivate the choice. It's also the case that most of the analyses drop a bunch of the pairs, only including pairs that have one bias and one no-bias stimulus in them. Why didn't simply only run those pairs? And tell us early on that you are dropping them, please.

Other comments:

Line 41: "are combined with weights proportional to their precision": this is tacitly assuming Gaussian noise if you are talking about a Bayesian model. You know this, but I like things explicit.

Lots of places: You randomize the reference orientation and avoid the cardinals. This effectively averages out any cardinal-direction priors. I was surprised you never discussed this explicitly.

Figure 1A: This is the only place \lambda is defined or used. For example, line 838 doesn't use \lambda.

144: "The strength of the likelihood" is a weird phrase. I'm not sure what aspect of the likelihood is meant (it's width???). Also, it's not completely controlled by \theta, because that's the stimulus orientation (relative to the reference), which differs from the noisy measurement, and it's the lateral upon which the likelihood is based.

157: Please indicate the prior direction \lambda = 35 for the blue and orange bias, either in the figure caption, in the main text (line 175), or both.

198: "intensity"???

282: In some ways this is the bottom line of the paper, but it's a pretty modest (12%) effect.

217: Why do you allow sensitivity to vary across conditions here, while using a constant sensitivity in the manipulation check task? Additionally, individual fitting results should be included in the appendix.

227: It actually makes sense that the choices do not match the estimates from the manipulation check task. In the manipulation check, the prior effect might be stronger because all trials in a staircase block are either biased or unbiased, whereas the main task trials are intermixed. At least the Methods' description of the manipulation check seems to imply the conditions aren't interleaved. Otherwise, clarify the Methods.

Figure 2B: The x-axis label here is incorrect (and the tick marks make that obvious). As in the running text, this is the difference of proportions first turned into z-scores.

Fig. 3A: The 2nd row is labeled "Stimulus", but the gray curve is a measurement distribution for a fixed stimulus.

284-301: A more detailed description of the CFC model is needed, at least in the appendix, as the parameter estimates are provided in the appendix without sufficient explanation of the model.

307: Maybe you should say 35 deg/ -145 deg?

408: "a single sample from the posterior": This doesn't seem feasible. The posteriors here are pretty complex, so how would an observer sample from it? And, the sample can be on the opposite side of the criterion compared to the participant's first-order response, which is weird. Why not use MAP or MEAN of the posterior instead? This is much simpler than sampling, and you have no evidence of sampling. Just because you don't ask for an estimate doesn't mean they don't infer one.

Figures 4 and later: These 4x4 colored figures: Are the data from pooling over participants? Didn't the participants get participant-specific stimuli chosen based on the control experiment? And wasn't that experiment unsuccessful in matching choice probabilities? As such, isn't it problematic to pool them in these categories, since the actual choice probabilities came out differently for different participants?

502: the "true motion direction" isn't known to the participant, so this vector projection should be based on a noisy measurement. Maybe the average will come out this way as plotted though in the expected value. Not immediately clear to me.

WPPCE model: This left a bad taste for me. You have a bunch of models. None work. But one is oriented vertically and one diagonally and the data look in between, so you took a weighted combination of failing models to get one that fit the data. Hmmmm. Here's another idea, perhaps equally distasteful, but that might make similar predictions: Have the prior depend on the task, as follows. The prior could be a Gaussian or a von Mises. Let the concentration parameter of the prior depend on \lambda, for example:

\kappa(\lambda) = \kappa_0 | \sin(2\lambda) |

This predicts increased bias in the bias condition because the prior not only shifts the posterior mean but also makes the posterior narrower, so there’s more probability density on one side of the criterion. In the no-bias condition, the posterior would be equivalent to the likelihood and wider than in the bias condition.

701: stimulus "values?". I presume you mean texture orientations, but please say so.

706 or so: State explicitly that the staircase controlled motion direction.

712: Why no lapse rate?

727: Why 3.5% here when it was 3% before?

751: 2.86 deg of visual angle IN LENGTH

765: This isn't the kind of stimulus for which I'd typically use Michelson contrast since most pixels were in the background. I'd report Weber contrast (i.e., 6%).

773: "cursor was invisible": Did it start out visible or did they know somehow that it started in the center? How else would they know where to move it?

782: I can't parse this sentence. What specific stimulus settings had no bias effect and how did you know that???

797: I think that, at least in Methods, you haven't yet said that you are omitting trials that aren't one each of bias/no-bias in the pair, but the wording requires that to be true.

816: into one psychometric FUNCTION

838: Should you state that these were circular Gaussians (that wrapped around) or their sigmas were low enough so that wraparound wasn't an issue?

892: It's weird to call this a "weight" when it's an SD and this isn't the Gaussian case so it doesn't end up acting (indirectly through precision) like a weight. I'll note, FWIW, that my lab has a paper (in press, Proc Royal Soc B) that claims that, in fact, people use an SD for inference that is smaller than the one we experimenters estimate from discrimination. But you folks already know that ;^)

915: There's no reason to include "radius" in this formula

Table S5: I'm glad you include fitted parameters (based on pooled data???). But, there are some big effects here (difference in confidence noise and in confidence boost) that you never discuss. Why not?

Reviewer #2: This paper investigates the role of long-term perceptual priors on confidence in perceptual decision making using a Bayesian framework. The authors manipulated their stimuli so that a natural prior favoring slow moving stimuli as explanations for percepts either informed perceptual decisions or was orthogonal to the decision at hand. They find that confidence is higher in decisions where the natural prior induced a perceptual bias. Based on this finding, the authors claim that confidence relies more on evidence that confirms existing beliefs (i.e., is informed by the natural prior). In my opinion this is an interesting study but its conclusions are not fully supported by the approach. The data and model need to be scrutinized further before I would find this acceptable for publication. I outline this below.

Major concern:

What the authors call the “Bayesian Optimal Observer Model” is too simplistic to draw the kinds of conclusions that the authors want to draw. The confidence in a judgement is directly related to the posterior probability of that judgement being correct. The posteriors of the bias and no-bias conditions are assumed to differ only because of the prior, while the likelihoods are assumed to be equal, but that is not properly motivated in my opinion. The long-term prior for lower speeds likely also biases the encoding and therefore affects the likelihood (e.g., https://doi.org/10.1038/nn.4105). The model formulation as presented assumes that the internal noise is independent of the angle between line orientation and line motion direction, but this is clearly not supported by previous data (https://doi.org/10.1167/11.3.25) and seems to be unsupported by the presented data. This suggests that the sensory measurements in the Bias condition were less noisy than the sensory measurements in the No-Bias condition, which naturally explains the difference in confidence.

This might be mitigated somewhat by the attempt to match the proportion of ‘blue’ and ‘orange’ choices in Bias and No-Bias conditions. Since different values of motion directions were used for the Bias and No-Bias conditions (line 185), it is not clear how big this difference in encoding precision between Bias and No-Bias conditions should be. The data however seem to show that participants were more sensitive in the bias condition (Fig 2), and they seem to have been more accurate at picking blue and orange when the stimuli were blue or orange respectively (lines 196-220). This is easily explained by varying the internal noise based on the angle between the line’s orientation and its motion direction.

Additionally, there are more sources of prior information available to the participant than just the orientation of the moving lines. The long-term prior that is manipulated by the authors (i.e., the line orientation) has to be combined with a shorter-term prior about the structure of the experiment. It is not immediately clear how this prior should be modeled, since participants received no feedback about their answers and so might have had uncalibrated assumptions about how correct their answers were. Whether the shape of this task prior is uniform or gaussian or something else is an open question. Without considering this task-related short-term prior, the “optimal Bayesian observer” is incomplete.

There is a chance that a proper ideal Bayesian observer model that includes all available prior information and an encoding asymmetry, where the precision of the likelihood varies with the angle between line orientation and motion direction, explains the data by itself.

The authors need to show either that a Optimal Bayesian Observer model that correctly accounts for the sensory noise does not reproduce the higher confidence in the Bias condition, or they need to show that the sensory noise (i.e., the precision of the likelihoods) does not change when the angle between line orientation and motion direction changes. Note that I don’t mean theta, the angle between reference and motion direction, but rather the angle between the line orientation and the motion direction, which is how the long-term prior is manipulated.

This seems conceptually similar to scaling the variance of the likelihood like in the Weighted Prior Confidence Model that the authors propose. However, in that model all Bias likelihoods get scaled by the same constant. If the sensory noise depended on the angle between line orientation and line motion direction, different conditions (e.g., P(“blue”)=0.15 and P(“blue”)=0.35) would be scaled differently. This difference should allow the authors to differentiate between the Optimal Bayesian Observer model and the Weighted Prior Confidence Model.

If it turns out that stimuli with more prior-congruent evidence lead to less variable measurements (as suggested by https://doi.org/10.1167/11.3.25), then the conclusions from this experiment should be very different from what the discussion presents at the moment. It would suggest that there is no confirmation bias in those confidence judgements at all.

Minor concerns:

- Some imprecise language should probably be cleaned up, for instance “strength of the likelihood” (line 134) probably means “precision of the likelihood”.

- Z-scoring of proportions that are bounded between 0 and 1 (lines 266-267) is not appropriate. The authors could use logit (log(p/(1-p))) or Fisher’s arctanh transformation instead or they might pick a different function to fit to the raw data, so that a transformation is unnecessary.

Reviewer #3: In this paper, the authors examined how long-term perceptual priors influence confidence judgments, using the slow-motion prior in motion perception. Participants were presented with pairs of tilted moving-line stimuli and made motion direction judgments (which the author categorized as blue vs. orange region) followed by a confidence forced-choice to decide which of the two intervals they were more confident in. The authors designed two conditions: a Bias condition, where the slow-motion prior systematically influenced perceived direction (thus favoring one response), and a No-Bias condition, where the prior had no directional influence. The key finding was that participants showed a confidence bias favoring the Bias condition, even after accounting for performance differences. Through computational modeling, they discovered this effect was best explained by a model where confidence uses both Bayesian posterior evidence (used in perceptual decisions) and prior-congruent evidence as an additional cue. They concluded that long-term perceptual priors, like previously studied high-level expectations, have a greater influence on metacognitive confidence than on perceptual decisions themselves.

The manuscript is overall well written, the experiments are carefully designed and well executed, and the topic addresses an important question in the field of perceptual decision-making and metacognition. The study contributes meaningfully to our understanding of how long-term priors shape confidence independently from decision accuracy and will likely be of interest to researchers working on Bayesian models of perception and confidence. However, I have several concerns regarding the modeling assumptions and analytical approach. These issues are detailed below.

Major comments

1. The authors implemented a Bayesian optimal observer model assuming that participants had perfect knowledge of the generative structure of the task, which would include the difficulty level in each condition (the “targeted fraction of choice”). However, it seems that participants were never explicitly informed of these conditions, and it is unclear whether they could have reliably inferred the statistical structure. Thus, given that the subjects can only infer the actual statistical state of the environment but not have a full accurate knowledge of it, I wonder whether it is fair to frame this as a test of the Bayesian Confidence Hypothesis (BCH) in its strictest sense (Li & Ma, 2020; Xue et al., 2024). Maybe it is worth to more carefully consider what task structure information was available to participants and discuss whether the model’s assumptions are reasonable representations of participants’ internal knowledge. In other words, the observed deviations from optimal Bayesian behavior might simply reflect incomplete knowledge rather than fundamentally different computational processes.

2. Another way I can think of to model the prior is to model it within a signal detection framework, and in such priors are typically modeled as a shift in the decision criterion (Green & Swets, 1966; Zheng et al., 2024). Under this framework, when making decisions aligned with the prior, subjects would naturally have higher confidence due to the greater distance between the stimulus evidence and the shifted criterion. Even when making a decision not aligned with the prior, it would not result in a higher confidence, like the prediction of the weighted prior model. This happened because the criterion shift means that the distance between the prior-congruent category and the decision criterion is larger. I wonder what the authors’ thoughts are on this.

3. It seems that even the best-performing WPPCE model shows noticeable discrepancies from the empirical data (as visible in Figures 7C & 7D). These remaining gaps might suggest that either important computational principles are still missing from the models or that the experimental paradigm introduces complexities not captured by the models. Further model exploration or a discussion of these limitations seems to be necessary.

Minor comments:

1. Did the authors examine the relationship between the strength of the prior and the strength of confidence bias across participants? It might also be interesting to see whether individual difference exists.

2. The paper would benefit from a more explicit discussion of how the findings relate to the broader Bayesian confidence hypothesis and Bayesian framework (Bowers & Davis, 2012; Griffiths et al., 2012).

3. The confidence models have different numbers of parameters, so AIC alone may not provide a complete picture. BIC results would offer complementary information for model comparison. At least consider adding it in the supplementary.

4. The experimental data heatmap only appears in Figures 4A and 7D, making it difficult to compare model predictions to the data. Maybe a difference maps between experimental data and model predictions when presenting results from different models would help?

5. A complementary way to examine the relationship between decision bias and confidence is through mixed-effects regression. One could regress confidence on false alarm rate (as a proxy for bias), accuracy, and RT. There is preliminary data suggesting that while bias positively correlates with confidence when considered alone, it negatively predicts confidence when accuracy is controlled, indicating that participants may metacognitively discount bias. Including such an analysis or discussing it would enhance the mechanistic understanding of the observed bias.

**Have the authors made all data and (if applicable) computational code underlying the findings in their manuscript fully available?**

Reviewer #1: Yes

Reviewer #2: Yes

Reviewer #3: Yes

PLOS authors have the option to publish the peer review history of their article (what does this mean? ). If published, this will include your full peer review and any attached files.

**Do you want your identity to be public for this peer review?** For information about this choice, including consent withdrawal, please see our Privacy Policy .

Reviewer #1: **Yes: ** Michael S Landy

Reviewer #2: No

Reviewer #3: No

**Figure resubmission:**

**Reproducibility:**

To enhance the reproducibility of your results, we recommend that authors of applicable studies deposit laboratory protocols in protocols.io, where a protocol can be assigned its own identifier (DOI) such that it can be cited independently in the future. Additionally, PLOS ONE offers an option to publish peer-reviewed clinical study protocols. Read more information on sharing protocols at https://plos.org/protocols?utm_medium=editorial-email&utm_source=authorletters&utm_campaign=protocols;

---

## [Decision Letter · Decision Letter 1]

15 Oct 2025

PCOMPBIOL-D-25-00503R1

Long-term perceptual priors drive confidence bias that favors prior-congruent evidence

PLOS Computational Biology

Dear Dr. Constant,

Thank you for submitting your manuscript to PLOS Computational Biology. After careful consideration, we feel that it has merit but does not fully meet PLOS Computational Biology's publication criteria as it currently stands. Therefore, we invite you to submit a revised version of the manuscript that addresses the points raised during the review process.

Please submit your revised manuscript within 30 days Dec 15 2025 11:59PM. If you will need more time than this to complete your revisions, please reply to this message or contact the journal office at ploscompbiol@plos.org. Please include the following items when submitting your revised manuscript:

We look forward to receiving your revised manuscript.

Kind regards,

Wolfgang Einhäuser

Academic Editor

PLOS Computational Biology

Lyle Graham

Section Editor

PLOS Computational Biology

**Additional Editor Comments :**

As you will see, the reviewers were very happy with your revision. There are some minor suggestions, which you should incorporate before the manuscript can be formally accepted.

**Reviewers' comments:**

Reviewer's Responses to Questions

Reviewer #1: Re-review of "Long-term perceptual priors drive confidence bias that favors prior-congruent evidence"

Reviewer: Michael Landy

This revision is much-improved with additional models and clarification and is pretty much ready for publication. I still have my reservations about the prior model (in particular about "confidence boost") and I still find the winning model a bit of a hack, averaging two models, one of which overpredicts and the other underpredicts. But, all in all, it's a clever project and quite interesting and thorough.

Very minor comments (numbering from the no-track-changes version):

Line 40: "assumed to have Gaussian noise" is a phrase that's applied to the likelihood and to the prior. I don't think of the width of the prior as "noise", since the prior is a fixed quantity (from the decision-maker's perspective), and thus not noisy. It's width is a component of knowledge of world statistics, but when used, it's just fixed.

428: The brain presumably DOES need to know the full distributional properties in order to carry out sampling, so this seems not quite true. I still don't love the sampling idea as a process model, don't know of prior papers that support posterior sampling for this and, personally, would prefer to see a posterior mean or mode model included or at least discussed.

459-461: You say you allow for a different prior width for estimation compared to for confidence, but then the model scales the likelihood rather than the prior. Sure, this is pretty much equivalent, but reads funny.

585: I'm happy about the addition of the efficient-coding model. Why do you need transduction noise? It's kind of a confidence-noise parameter, which the other models don't require. Why can't it account for confidence bias when the model predictions look pretty good?

Four-panel confidence plots (several figures): Please make the y-axis scales and tick mark labels the same for all such figures.

Reviewer #2: The authors did a great job addressing my concerns, particularly through the addition of the Efficient Coding model.

I have some remaining minor suggestions for the revised manuscript:

The new figure panel style for the model predictions (e.g., figure 3D and 4B etc) provide valuable information, but I found them somewhat difficult to parse. If possible, simplifying these figures or spreading the information across additional panels might improve readability. Maybe removing the gray background will help too. It’s also possible that one could flip one of the Colors to combine the data (i.e., turn every Blue into an Orange by changing the sign on stimuli and answer), so that some panels and lines can be eliminated but I am not entirely certain that this is balanced.

Please clarify in the methods section whether the models were fitted exclusively to the perceptual choice data, exclusively to the confidence judgment data, or jointly to both.

It seems to me that another possible interpretation of the WPPCE model is that instead of an “overweighted” prior, the observers are working with a less precise likelihood (i.e., the sensory evidence has degraded in working memory). Since posteriors are normalized, both overweighting the prior and underweighting the likelihood have the same effect on the subjective probability of being correct. I would appreciate it if the authors could briefly comment on this.

Reviewer #3: Thank the authors for addressing my comments. The paper looks all good now. Just one remaining thought: the authors mentioned in the reply that in their task there is no explicit sensory criterion because the reference moved from trial to trial, but my impression is that even though the reference varied, each trial still effectively defined a decision boundary. For example, the confidence could, in principle, depend on the distance between the perceived motion direction and this trial-specific criterion. I understand that the author might want to maintain a Bayesian generative framing, but it would be great to see what the authors think about it.

Also, some formulas are not properly displayed (line 300 – 308).

**Have the authors made all data and (if applicable) computational code underlying the findings in their manuscript fully available?**

Reviewer #1: Yes

Reviewer #2: Yes

Reviewer #3: Yes

PLOS authors have the option to publish the peer review history of their article (what does this mean? ). If published, this will include your full peer review and any attached files.

**Do you want your identity to be public for this peer review?** For information about this choice, including consent withdrawal, please see our Privacy Policy .

Reviewer #1: **Yes: ** Michael S. Landy

Reviewer #2: No

Reviewer #3: No

**Figure resubmission:**
---

## [Editor Report · Decision Letter 2]

9 Dec 2025

Dear Ms. Constant,

We are pleased to inform you that your manuscript 'Long-term perceptual priors drive confidence bias that favors prior-congruent evidence' has been provisionally accepted for publication in PLOS Computational Biology.

Best regards,

Wolfgang Einhäuser

Academic Editor

PLOS Computational Biology

Lyle Graham

Section Editor

PLOS Computational Biology

---

## [Editor Report · Acceptance letter]

PCOMPBIOL-D-25-00503R2

Long-term perceptual priors drive confidence bias that favors prior-congruent evidence

Dear Dr Constant,

I am pleased to inform you that your manuscript has been formally accepted for publication in PLOS Computational Biology. Your manuscript is now with our production department and you will be notified of the publication date in due course.

With kind regards,

Judit Kozma
